# Video Packet Distribution Scheme for Multimedia Streaming Services in VANETs

**DOI:** 10.3390/s21217368

**Published:** 2021-11-05

**Authors:** Yongje Shin, Hyunseok Choi, Youngju Nam, Euisin Lee

**Affiliations:** School of Information and Communication Engineering, Chungbuk National University, Cheongju 28644, Korea; yjshin@cbnu.ac.kr (Y.S.); plazpt@cbnu.ac.kr (H.C.); imnyj@cbnu.ac.kr (Y.N.)

**Keywords:** vehicular ad-hoc network, multimedia streaming service mobility support, trajectory, packet distribution, vehicle mobility

## Abstract

By leveraging the development of mobile communication technologies and due to the increased capabilities of mobile devices, mobile multimedia services have gained prominence for supporting high-quality video streaming services. In vehicular ad-hoc networks (VANETs), high-quality video streaming services are focused on providing safety and infotainment applications to vehicles on the roads. Video streaming data require elastic and continuous video packet distributions to vehicles to present interactive real-time views of meaningful scenarios on the road. However, the high mobility of vehicles is one of the fundamental and important challenging issues for video streaming services in VANETs. Nevertheless, previous studies neither dealt with suitable data caching for supporting the mobility of vehicles nor provided appropriate seamless packet forwarding for ensuring the quality of service (QoS) and quality of experience (QoE) of real-time video streaming services. To address this problem, this paper proposes a video packet distribution scheme named Clone, which integrates vehicle-to-vehicle and vehicle-to-infrastructure communications to disseminate video packets for video streaming services in VANETs. First, an indicator called current network quality information (CNQI) is defined to measure the feature of data forwarding of each node to its neighbor nodes in terms of data delivery ratio and delay. Based on the CNQI value of each node and the trajectory of the destination vehicle, access points called clones are selected to cache video data packets from data sources. Subsequently, packet distribution optimization is conducted to determine the number of video packets to cache in each clone. Finally, data delivery synchronization is established to support seamless streaming data delivery from a clone to the destination vehicle. The experimental results show that the proposed scheme achieves high-quality video streaming services in terms of QoS and QoE compared with existing schemes.

## 1. Introduction

With the rapid development of ad-hoc wireless communications and vehicular technologies, vehicular ad hoc networks (VANETs) have enabled the delivery of data between vehicles on roads [1]. In VANETs, vehicles adopt the roles of network nodes owing to their high mobility during communication. Roadside units (RSUs) are basically installed near roads and act as relay nodes because of their static locations. Both vehicles and RSUs can communicate with each other via the dedicated short-range communication (DSRC) technology [2] for vehicle-to-vehicle (V2V) and vehicle-to-infrastructure (V2I) communications [3,4]. Many projects (e.g., VICS [5], CarTALK 2000 [6], and Network-on-wheels (NoW)) [7]) and industry groups (e.g., Car2Car Communication Consortium [8]) have conducted various studies to establish intelligent transport systems using VANETs. In the intelligent transport systems, VANETs offer drivers and passengers safety and convenience, and furthermore support entertainment and environment monitoring applications [9]. Many studies on VANETs have addressed various applications, such as car accident warning for safe driving, emergency vehicle access for public service, road congestion notice for improved driving, and commercial advertisement for business [10,11,12].

### 1.1. The Challenges of the Video Streaming Services

Among the applications of VANETs, video streaming services are currently receiving much attention for supporting applications such as video warning for safe driving and video chats for entertainment. A vehicle can send a video warning file about an accident on its forward path to the vehicles behind it. It can also share a video clip file advertising nearby stores with its neighboring vehicles through a video chat application. Generally, video files contain large amounts of data depending on their related applications and have stringent quality of service (QoS) and quality of experience (QoE) limitations. VANETs have dynamic topology features owing to the intermittent connections and frequent link breakages due to the high mobility of vehicles on roads. Accordingly, three challenging issues should be addressed to achieve the QoS and QoE requirements to support video streaming services in VANETs with dynamic topology features and intermittent connections. First, large video streaming data should be divided into many data chunks, which should be sequentially delivered along the travel route of the destination vehicle. Second, the video data chunks should be forwarded via paths with good link quality to ensure their QoS and QoE requirements. Third, data caching should be efficiently conducted to store the video data chunks on the travel route until the destination vehicle arrives.

### 1.2. Background Studies

Until now, numerous studies have proposed protocols to deliver data to a vehicle in VANETs [13]. They are broadly categorized into two approaches: topology-based [14] and position-based [15] routing. Generally, many studies have shown that a position-based routing approach has potential for VANETs because it performs well with mobile nodes, such as vehicles [15]. Topology-based routing protocols, such as ad-hoc on demand distance vector (AODV) [16] and destination sequenced distance vector (DSDV) [17], depend on the topology information of the network to construct routing paths. Thus, when vehicles continuously move on roads, the topology information frequently changes, and consequently, the routing paths are broken. On the other hand, position-based routing can forward a packet to its destination node only using the position information of the nodes in a forwarding decision. Position-based routing protocols, such as greedy perimeter stateless routing (GPSR) [18] and greedy perimeter coordinator routing (GPCR) [19], use a greedy forwarding method to forward packets to the nodes that are progressively closer to their destination. To overcome the problem of path failures due to the high mobility of vehicles in VANETs, junction-based multipath source routing (JMSR) [20] exploits multiple paths and a backbone-assisted hop greedy (BAHG) method [21] exploits the best path based on the connectivity status. To achieve efficient data delivery by exploiting the mobility information of vehicles on roads, trajectory-based data forwarding (TBD) [22] and trajectory-based statistical forwarding (TSF) [23] propose trajectory-based V2I and infrastructure to vehicle (I2V) multi-hop routing protocols, respectively. DTI [24] presents a data delivery and caching scheme using a rendezvous point between the trajectory of a destination vehicle and the routing path of the data based on the road map information. However, because the existing schemes [18,19,20,21,22,23,24] do not support an elastic and continuous video packet delivery to a destination vehicle, the mentioned protocols are not suitable for video streaming services in VANETs.

For supporting multimedia streaming services in VANETs, many protocols [25,26,27] have been proposed using location information. GPSR-2P [25] is a routing protocol based on GPSR for video transmissions. LIAITHON [26] identifies multiple short paths based on location information for reliability. VIRTUS [27] uses the contact time between neighbor vehicles based on their future locations to select good relaying vehicles. Because both QoS and QoE are the main requirements for users to be satisfied with video streaming services in VANETs, many protocols [28,29,30,31,32] have been proposed to support them. CBQoS-Vanet [33] and QoScompliant [28] use QoS metrics, such as bandwidth and end-to-end delay, to determine the best route for video streaming data. DQLTV [29] and QORE [31] select the best relay node to forward video streaming data based on video-related QoE parameters, such as frame loss and jitters. To support both QoS and QoE, DBD [32] uses a backbone with persistent and high-quality routes, whereas GeoQoE [30] employs QoE-aware geographic routing based on the QoE value calculated by the correlated QoE and QoS factors. However, these protocols [28,29,30,31,32] for supporting the QoS and QoE of video streaming data do not consider the delivery of multimedia streaming data to a moving destination vehicle. Some protocols [34,35,36] have been proposed to support multimedia streaming services by using the mobility information of vehicles. 3MRP [34] and an adaptive video uploading scheme [35] select the best relay access point (AP) or vehicle for delivering video streaming data based on the mobility information of vehicles. Zhao et al. [36] caches over-the-top (OTT) multimedia streaming content in the future connected RSUs by the mobility prediction of vehicles. However, they did not consider the trajectory information of a moving destination vehicle to deliver or cache multimedia streaming data. In summary, all of these previous protocols [25,27,29,30,31,32,34,35,37] on multimedia streaming services cannot properly handle the three challenging issues to provide video streaming services with supporting both QoS and QoE to destination vehicles in VANETs.

### 1.3. Contributions

Therefore, we propose a video packet distribution scheme named Clone, which supports video streaming services with QoS and QoE from a source node to a destination vehicle in VANETs. To achieve this, the proposed scheme sequentially and seamlessly distributes video streaming packets to APs called clones on the trajectory of the destination vehicle by integrating high quality V2V and V2I communications in order to deliver video streaming data from the source to the destination vehicle. To realize the proposed scheme, we first define an indicator called current network quality information (CNQI). It measures the feature of data forwarding of each node to its neighboring nodes in terms of both reliability and throughput, which impact the data delivery ratio and the delay related to the QoS factors for delivering video streaming data, respectively. The proposed scheme chooses vehicles with high CNQI values as the relay nodes for delivering the video streaming packets. Next, we decide clones among nodes on the trajectory of the destination vehicle to cache the video streaming packets from the source for the destination vehicle. The clones are selected based on the CNQI values and the data delivery ability of nodes to improve the QoS. The number of clones is determined based on the size of the video streaming data and the data storage size of nodes. Next, we provide a packet distribution optimization to determine the maximum number of video packets to cache for the destination vehicle in each clone and to allow sequential video packet delivery to achieve better QoE. The determination is based on the resource ability parameters related to data caching and forwarding in each clone. Finally, we provide data delivery synchronization, enabling each clone to provide seamless forwarding of its video streaming packets to the destination vehicle in its communication coverage in order to achieve high QoE without any buffering effects on the video streaming. The synchronization ensures packet calibration for chronological order forwarding based on the delay calculation for packet loss reduction and the mobility estimation for the destination vehicle. To evaluate the performance of the proposed scheme, we compare it with four existing schemes by simulations: TSF, which is the basic trajectory based single packet transmission scheme; BAHG, which is the enhanced version of the TSF that considers vehicle’s trajectory to predict the intersections passing by; DQLTV, which is the video streaming service that supports QoE only; and GeoQoE, which is the video streaming services that supports QoE and QoS with geographical scheme. All schemes are implemented using the NS-3 [38] simulator to emulate video streaming services with a large amount of data to vehicles on the Seattle city map. The simulation results obtained in various environments verify that the proposed scheme achieves high-quality video streaming services with shorter end-to-end delay and higher frame delivery ratio in terms of the QoS and the QoE compared to the four existing schemes.

The remainder of this paper is organized as follows. First, we present the related work of our Clone scheme in Section 2, in which we address and categorize protocols from general packet transmissions to video packet transmissions for multimedia streaming services in VANETs. Section 3 describes our Clone scheme in detail in terms of its four phases. Simulation results are provided to evaluate the performance of our Clone scheme in Section 4. Finally, the conclusions of this paper are provided in Section 5.

## 2. Related Works

The objective of this study is to route video streaming data to a vehicle moving toward its destination according to its trajectory in a VANET. In this section, we review the studies related to our study aim. First, we first examine the studies on data delivery to a destination in VANETs. Numerous routing protocols have been proposed to support data delivery to a destination vehicle in VANETs. Generally, they are categorized into two routing approaches in VANETs: topology- and position-based routing. Topology-based routing protocols, such as AODV [16], DSDV [17], and dynamic source routing (DSR) [39], are most suited for mobile ad-hoc networks (MANETs). However, they are known to fail reaching a satisfactory performance for urban VANETs. Topology-based routing relies on the topology information of the network to construct the routing paths. However, in VANETs, because vehicles continuously move on roads, the topology information frequently changes, and thus, the routing paths are broken. Consequently, data delivery cannot be achieved.

The second category is the position-based routing approach. It exploits only the pure position information of nodes, instead of the topology information, to route data to destinations. Thus, it is considered as a promising routing approach in the dynamic environments of VANETs. In position-based routing, a source node encapsulates the position of the destination node in each data packet and subsequently forwards the packet to one of the one-hop neighbor nodes toward the position. The neighbor node receiving the data packet is geographically closest to the position of the destination node. This process is called greedy forwarding in geographic routing. Early position-based routing protocols only used greedy forwarding, which cannot prevent frequent occurrence of local maximum traps. A local maximum trap is a state in which a vehicle cannot find any neighboring vehicle closer to the destination than itself. GPSR and GPCR are two well-known protocols of this category, which both move out of local traps by complex perimeter routing. In [14,15], it was shown that position-based routing protocols that do not use the trajectory information of vehicles have unacceptable performance for urban VANETs. To overcome this issue, routing protocols using a digital map force the packets to be transferred just inside the framework of streets and junctions for the route of data delivery in VANETS. The routes identified by these routing protocols consist of names of streets and junctions. To achieve this, geographic source routing (GSR) [40] was proposed as a map-based routing. However, it lacks connectivity estimation, and thus, has a low data delivery ratio. To increase the data delivery ratio, JMSR [20] exploits two independent routes and a path repair strategy. In comparison, the BAHG [21] method is based on the usage of backbone vehicles and a greedy forwarding method. The route between the source and the destination vehicles is identified by a hop-greedy procedure, which derives Dijekstra and greedy routing mechanisms. Although the position-based routing protocols guarantee data delivery to destinations, they are inefficient for supporting destination vehicles moving according to their trajectories in VANETs.

Because a vehicle generally moves according to its trajectory toward its destination, routing protocols [22,23,24] exploiting the trajectory information of vehicles have been proposed to achieve efficient data delivery in VANETs. They calculate the trajectory of a destination vehicle and forward data packets on the trajectory. Subsequently, the destination vehicle receives the data packets while moving along the trajectory. Thus, they achieve efficient data delivery to a vehicle moving on its trajectory. Jeong et al. proposed a protocol called TBD [22] for a V2I communication study. TBD obtains the trajectory information of a vehicle from a localized system, such as on-board navigation units. It also calculates the road expected delay of a road segment based on the computed end-to-end delays among the network devices. Based on the trajectory of vehicles and their position information, TBD also determines the end-to-end expected data delivery delay. In addition, the vehicles exchange the computed delay with other neighboring vehicles in their range to determine the best next-hop vehicle on the street. Thus, by combining the trajectory information with vehicular traffic statistics, TBD achieves reduced end-to-end expected data delivery delay from a vehicle to an Internet AP. Jeong et al. also proposed a protocol called TSF [23], which is an extension of TBD for supporting the I2V data delivery from an AP to a vehicle. TSF uses multi-hop communications to a selected target point on the trajectory of the vehicle. Subsequently, the data delivery uses the trajectory of the vehicle that passes the target point. The target point is an optimal rendezvous point of the data packet and the destination vehicle. It is selected optimally for achieving excellent delay performance. Shin et al. proposed a data delivery protocol using the trajectory Information on a road map in VANETs (DTI) [24]. TBD and TSF are unable to decide the optimal reception point in a high mobility situation. Thus, DTI uses the optimal reception point on the trajectory of a destination vehicle by employing an arithmetical model based on road map information for advantageous cost-minimized data delivery. Moreover, DTI contains a data caching algorithm to deal with the trajectory changes of a destination vehicle due to its geographical changes. However, because a large amount of video streaming data demands an elastic and continuous video packet delivery approach to a destination vehicle, if using the existing schemes it might be difficult to support video streaming services. This is owing to the feature that a destination vehicle continuously changes its locations owing to its high mobility in VANETs.

For satisfying the demands of multimedia streaming services and applications in VANETs, many routing schemes [25,26,27,37,41] have been proposed to deliver multimedia streaming data. To support video streaming distributions, Soldo et al. [37] proposed a streaming urban video (SUV) protocol based on a distribution structure built over the physical topology of vehicles called relay nodes. Because an SUV is based on time division multiple access (TDMA), the relay nodes have to follow a scheduled time to access the channel to distribute the video data. VIRTUS [27] exploits future locations of vehicles to predict the contact time between neighbor vehicles to select relaying vehicles. Location-aware multipath video streaming (LIAITHON) [26] identifies relatively multiple short paths with minimum route coupling effect based on location information for video transmission. [41] is a three-path LIAITHON enhancement, such that each path includes a wait time calculation scheme and a route coupling prevention scheme. GPSR-2P [25] is a routing protocol based on GPSR for video transmissions in urban VANETs. To overcome the delay and packet loss problems and avoid congestion and saturation in a single path, GPSR-2P considers the first two nearest neighbors toward the destination to find multiple paths for delivering video packets for streaming services. However, even though the above-mentioned studies [25,26,27,37,41] conducted research on multimedia streaming applications and services, they did not consider the QoS and the QoE, which are the main factors affecting user satisfaction for video streaming services.

Thus, many routing protocols have been proposed to support the QoS and QoE of video streaming services in VANETs. For supporting the QoS requirements of multimedia services, Lakas et al. [28] proposed CBQoS-Vanet with a bee colony inspired algorithm that calculates the best routes from a source to a destination based on QoS metrics, such as bandwidth, end-to-end delay, and jitter. Xing et al. [42] proposed a hybrid framework to determine the best delivery strategy and select an optimal path for multimedia data dissemination by considering the delivery delay, storage cost, and QoS. To enhance the QoE of video streaming, DQLTV [29] selects forwarding vehicles with the best link quality (transmission success) and link availability (lifetime) using the mobility information of vehicles. QORE [31] applies geographic statistical routing and uses QoE-aware and video-related parameters for maintaining a reliable multimedia data delivery backbone and selecting the best relay node on the backbone. To guarantee a reasonable trade-off between video quality and video buffer time in HTTP-based streaming systems, Bokani et al. [43] employed the Markov decision process to select the optimum chunk under the dynamic channels to achieve QoE of video streaming. To provide QoE as well as QoS of video streaming, DBD [32] exploits a backbone-based approach to create and maintain persistent and high-quality routes to support the delivery of video data through V2V communications. GeoQoE [30] conducts QoE-aware geographic routing for video streaming, which measures the QoE values (named mean opinion score (MOS)) of neighboring vehicles based on correlated QoE and QoS factors (e.g., packet loss rate, jitters, and delay). Subsequently, it selects one with the best QoE value as the relay vehicle. Using the mobility information of vehicles, some papers have proposed methods to efficiently support the QoS and QoE of multimedia streaming services in VANETs. The method, 3MRP, exploits the trajectory information of vehicles as one of five QoS metrics (distance to destination, vehicle density, trajectory, available bandwidth estimation, and MAC layer losses). Accordingly, 3MRP [34] enables a vehicle to select the best forwarding node for sending video reporting messages to an AP in the infrastructure of the city to alert emergency services. By exploiting the mobility prediction of vehicles, an adaptive video uploading scheme [35] was proposed to reliably deliver video streaming data from a moving vehicle to a fixed network by selecting optimal APs and stable relay vehicles. LBP [44] is a prediction window-based video streaming algorithm to adjust the requested QoE of a video while ensuring no video stall occurs for vehicles. In addition, by supporting QoS and QoE in vehicular streaming services, some cache-enabled streaming protocols have been studied in wireless networks. To enhance the QoE of video streaming, Pederson et al. [45] proposed an algorithm to support the adaptive bit rate (ABR) based on video characteristics while efficiently using caching in radio access networks. Yashuang et al. [46] proposed a dual time-scale dynamic cache scheme in base stations for supporting ABR streaming under the condition of high channel variations to achieve high QoS and QoE of video streaming services in VANETs. Zhao et al. [36] proposed a scheme to cache OTT multimedia streaming content in the future connected RSUs using the mobility prediction of vehicles. However, the existing protocols supporting the QoS and QoE of multimedia streaming services only consider static destination vehicles. Moreover, they exploit the mobility prediction and caching with the mobility information of normal vehicles on roads, instead of the trajectory information of the destination vehicles.

As we have examined before, in Table 1 we categorized the mentioned protocols of the related work. The existing protocols proposed for data delivery in VANETs are different from our Clone scheme. The proposed scheme considers the delivery of multimedia streaming data from a source to a destination vehicle that moves along its trajectory on roads. Subsequently, our Clone scheme supports both QoS and QoE for the delivery of multimedia streaming data to a destination vehicle with real-time (RT) support. In our Clone scheme, the QoS and QoE support is achieved by constructing both a caching storage on the trajectory of the destination vehicle and a reliable route to the caching storage, which are determined by our CNQI indicator based on the reliability and the throughput. Table 1 shows the summary of the related works on video streaming services in VANETs.

## 3. The Proposed Scheme

In this section, we present the proposed scheme for supporting video streaming services in VANETs. First, we explain the network model and the overview of our Clone scheme. Subsequently, we describe in detail the four phases composing our Clone scheme. Figure 1 shows the overall description drawing of the Clone streaming service. The figure explains the main processes of the proposed scheme: Clone node selection, video data distribution, and packet delivery synchronization. Figure shows the overview of the proposed protocol with the flow arrows. From the backbone server, which is connected to the internet service, the requested video contents file is distributed and forwarded to RSU. The distributed video files are delivered through the backhaul link to RSU, and the RSU forwards the received video files to the Clone nodes through the calculated path using APs. The Clone nodes are selected based on the user vehicle’s trajectory to support the vehicle’s mobility. Hence, the Clone nodes forward the distributed video files in time order while the user vehicles move towards the destination.

### 3.1. Network Model and Overview

VANETs pose a demanding challenge for the fulfillment of the stringent requirements for video streaming using the mentioned communication technologies. The main challenges are a result of the high dynamic topology, extensive coverage problem, and intensely varying density of VANETs. In addition to these challenges, any video streaming solution must comply with some QoS requirements. CISCO has defined some requirements for general video streaming. The delay should not be longer than 4–5 s, and the loss should not exceed 5%. Bandwidth requirements depend on applications, and jitter imposes no significant requirements. The transmission of a video demands the use of significant network resources; however, it cannot be excessive. Therefore, video streaming solutions for VANETs have to fulfill all these requirements limited to a reasonable occupation of the wireless medium.

The proposed scheme aims to minimize the communication delay between the user vehicles and the APs, having the best quality based on the CNQI metrics. Furthermore, the proposed scheme has a process to select the best nodes in the topology and uses a video packet distribution algorithm for a seamless streaming service to the user vehicles. Distributed nodes in the topology are fixed with a random position and the nodes density maintains the specific rate. Additionally, the clone node located at the end of the path to a user vehicle has a link stickiness process to forward the distributed video packets. Before the scenario, numerical integration of all communication nodes is inevitable to reduce the overall delay. Moreover, the integration can be a part of the proposed scheme. Each node maintains updated CNQI values based on a connection performance test along with the nodes of the neighbors in the one-hop range. The connection performance test focuses on enhancing the QoS values, such as bandwidth, bottleneck inflation in devices, and error rates.

The proposed scheme can be divided into CNQI measurement, clone node selection, packet distribution optimization, and synchronization phases. In the CNQI measurement phase, the CNQI value is calculated, which unifies the resource state and the availability of the nodes in the network into a single numerical value by a mathematical formula. The clone node selection phase selects relay nodes as the clone nodes for the proposed scheme. In this phase, the number of nodes and the locations of the nodes are determined based on the calculated CNQI information. Using the decided number of clone nodes, the proposed scheme initiates the packet distribution optimization phase. In this phase, the packets for transmission to a destination are distributed to all clone nodes for real-time seamless high packet forwarding. In the synchronization phase, a one-hop link is connected along with the selected clone nodes and the user vehicle. It utilizes the user trajectory information for a seamless link connection between the last and following link connection times of the clone nodes. Thus, the proposed scheme conducts the phases mentioned above in sequence to provide a high-packet data content service through communication between the selected clone nodes and the user vehicle.

### 3.2. Current Network Quality Information

In previous studies, many search techniques have been proposed for a single packet or small data transmission. However, the existing node information measurements are unsuitable for live video streaming services, owing to the low reflect rate of real-time node scenarios. Real-time resource availability information is crucial for live streaming support for calculating the optimal forwarding path to a service user with high-quality load balancing. The CNQI is an indicator of calculating the network efficiency of the overall nodes, which affects the user QoS and QoE of the overall topology. Several factors affect the video streaming service performance of the protocol. The geographical factor affects the error rate of a node, and it varies with moving obstacles. It is difficult to calculate and satisfy user QoS if the real-time topology is not considered. The video streaming data that a user receives have a quality status. If the transmitted video data have noise and delay when a user plays them, the user QoS will be decreased. Therefore, analyzing the overall topology by the CNQI technique is inevitable for user QoS and QoE. Using the CNQI scores of the nodes, we can determine which nodes cannot forward the multimedia data with a high MOS and SSIM score owing to obstacles and overload. The CNQI provides the information from reliable data, such as bandwidth of a node, current usage of a node, available resource by hardware score, and neighbor distribution rate in the range. Each node broadcasts a packet to estimate the quality of the video transmission in the one-hop range. The packet header includes the MOS and SSIM quality test data about the QoS and the QoE; then they compute their CNQI score and share each other. Finally, all nodes in the topology share the CNQI scores for multimedia transmission.

We modify the factors in the CNQI equation and describe them in detail below. *H* represents the performance of a node in case of hardware, and it ranges from 1 to 10 based on the response time and the hardware score. A single packet transmission test can calculate the hardware score in the communication range of a AP to the destination vehicle without any obstacles. *B* is the bandwidth of the AP, and the bandwidth usage is calculated by the number $N_{Users}$ of users that connected the AP for receiving video chunks. *E* represents the error rates due to the geographic feature and network topology of the node. $U_{Net}$ represents real-time network usage. The interaction of the network topology with neighbors, traffic information, and source data size defines the real-time network usage Based on these factors, the CNQI score can be calculated with Equation (Equation 2).
(1)$$U_{Net}=\frac{B}{N_{Users}},$$
(2)$$CNQI=H\times \left(1-E\right)\left(\frac{1}{U_{Net}}\right).$$

A high CNQI value shows the efficient network performance of a node when the node forwards packets in the networks. Furthermore, using the CNQI value, the proposed scheme has a traffic distribution effect on the overall usage of the nodes. In addition, the proposed scheme chooses relay nodes using the CNQI values in the forwarding phase, and because a CNQI value shifts with the allocated user number in the nodes, node accessibility can be improved. It is also possible to prevent a scenario where a sizeable geographic advantage causes numerous users to collect much traffic on one node, thereby degrading the efficiency of the node. In the clone node selection phase, all nodes keep updating the data table, which contains real-time information based on the assumption that all nodes keep updating their CNQI information. Because the table contains the information on variable factors, we can determine the status of the nodes in real time for realizing live streaming service. When the nodes have received the neighbor node searching for packets, the proposed scheme identifies the potential relay nodes because the nodes reflect the CNQI information.

### 3.3. Clone Node Selection

In the clone node selection phase, the best quality nodes located on the trajectory of the user vehicle are chosen. A clone node is a relay node that maintains synchronization with the user vehicle toward the destination. In the topology, many nodes participate in multimedia transmission. All relay nodes have a different capability and availability for communication owing to various reasons, such as hardware, network status, and topology status of the neighbor nodes. A clone node is selected for its high QoS and QoE for the user vehicle. It is an end node through which the multimedia packet passes to reach the user vehicle. Therefore, choosing the best clone node for the proposed scheme is the best approach to improve transmission efficiency. The CNQI score of a clone node must be higher than those of any nearby relay nodes because significant tasks are assigned to a clone node. A clone nodes has two primary roles, which are explained below: caching the multimedia contents for the user vehicle and maintaining synchronization with the user vehicle.

Clone node selection is the process of determining the number of relay nodes that a user will use for receiving the data. The first relay node forwards simultaneously content searching messages and one-hop neighbors CNQI searching messages. After the first relay node selection, the N-th node receives the user request content messages, including the content information. Subsequently, the proposed scheme determines the number of N-th nodes that will be used based on the trajectory of the user, CNQI value, content information, and topology status. If the size of the content is sufficiently large to be completed by one relay node, this process is omitted.

Seamless connection of the clone nodes with a user vehicle that has high mobility is a challenging issue. For this, the link connection of the previous clone node with the user vehicle must be connected until the next clone node connects with the user vehicle. The proposed scheme estimates overlapping geographical areas among the clone nodes located on the user vehicle trajectory. The overlapping geographical areas are estimated by the communication range of the clone nodes. If the overlapping area is insufficient to provide the transmission time, link disconnection may occur. However, if the overlapping area is extremely large, the number of clone nodes in the proposed scheme will be increased, leading to bandwidth dissipation. Thus, the overlapping area has to be optimized by the escape time of the user vehicle. Euclidean distance calculation applied for the optimization uses the entry and escape points of the overlapping area. From the optimization area, the guard band for distributed packet forwarding can be calculated. The guard band prevents QoS degradation caused by packet corruption and unstable transmission service owing to vehicle mobility.
(3)$$G_{min}=\frac{P_{out}^{k}\cdot V_{min}}{P_{{in}^{k+1}}\cdot V_{s}\cdot r_{play}}\leq G$$

The minimum guard band $G_{min}$ is the threshold and the minimum requirements to prevent link failure due to the lack of link changing time. $P_{out}^{k}$ represents the communication speed when the user vehicle leaves out from the $Clone_{k}$. $V_{min}$ represents the minimum time of the user vehicle stays in the overlapping area among the $Clone_{k}$ and $Clone_{k+1}$. $P_{in}^{k+1}$ represents the communication speed when the user vehicle enters the $Clone_{k+1}$. $V_{s}$ represents the time that the user vehicle stays in the overlapping area. $r_{play}$ represents the bit-rate of the video file. Note that when using Equation (Equation 3), the approximate guard band size is obtained to optimize the number of clone nodes for content data transmission. The formula sequentially assigns high-efficiency nodes using the calculated data based on the size of the remaining content data to be delivered and the CNQI values of the nodes on the user vehicle trajectory. If the content data to be transmitted must use two or more relay nodes, the data must be distributed to the relay nodes. The packet distribution optimization method is explained subsequently in Section 3.4.

At the end of the relay nodes, the last node is called a clone node, which replicates the network information of user vehicle for link connection. The clone node has a replicated identity (ID) from the user vehicle. The content data of the source identify the clone node as a user vehicle for convenience data forwarding in turn. A clone node must meet certain requirements, which are summarized as follows:Considering the QoS and QoE effectiveness of the proposed scheme, a clone node should be located on the trajectory of the user vehicle with a wide-open area that leads to minimum obstacles for communication.Based on the guard band equations and the CNQI for improving the QoS and QoE, all nodes should have the possibility of being relay and clone nodes, and a clone node must have extensive content storage for link connection and caching the algorithm.We designed a packet distribution algorithm and a synchronization algorithm for link connection based on the mobility of the vehicle and the content playtime. For the algorithms, a clone node must have an AP to receive the computed data.

### 3.4. Packet Distribution Optimization

In large content data transmission cases, one replicated clone node with a few relay nodes cannot forward all content data to a user vehicle owing to the size of the content compared to the storage in the nodes and the mobility of the user vehicle. Consequently, the user vehicle does not have sufficient time to receive the content data. We proposed a clone node selection algorithm to solve this problem by reasonably selecting the relay nodes as clone nodes based on the CNQI. Furthermore, the content must be distributed over the relay and clone nodes owing to extensiveness of the content data in video streaming transmission. Therefore, we suggested packet distribution optimization of the received data. The CNQI information from the user vehicle, such as trajectory, speed, and request content size, is used to select the clone nodes. In addition, the information of the relay node is crucial for choosing the clone nodes. The information obtained from the localization devices added to the user vehicle is variable, owing to the geographic distribution of the nodes.

After the clone nodes are selected based on the mobility of the user vehicle, node distribution, bandwidth usage of the nodes, and other information, the source node, which contains the contents, distributes the content data and subsequently forwards them to each clone node. All clone nodes receive the distributed packets respectively in order by time, which depends on the synchronization time of the user vehicle with each clone node and the time computed when the source node requests neighbor nodes on the user vehicle trajectory. To process the packet distribution optimization, we denote $N_{max}$ as the maximum packet forwarding amount of the clone node to the user vehicle and $V_{\left(n\right)}$ as the number of vehicles using the clone node in real-time. $D_{user}$ is the remaining distance of the user vehicle to out of the node communication range, $S_{user}$ is the speed of the user vehicle, ρ is a speed correction value for facilitating packet calculation of the speed to have in the future by mobility prediction of the user vehicle, and Guardband is the calculated range of the guard band size from the previous equation. In packet transmission, σ is the size of bits per chunk, which is the packet amount for 3 s of video playtime.

Using the factors we mentioned above, we can calculate the packet amount for the transmission to the user vehicle from each clone node. The content data are distributed and forwarded to each node based on the equation. The amount, $I_{{Clone}_{k}}$ is allocated data size of $Clone_{k}$. The distributed packets are allocated to the Clone nodes. $Clone_{k}$, is expressed in Equation (Equation 4).
(4)$$I_{{Clone}_{k}}=\frac{Guardband\times D_{user}}{\frac{\rho +S_{user}}{\sigma }}÷\left(1-\frac{N_{max}}{V_{\left(n\right)}\cdot \sigma }\right)$$

Based on Equation (Equation 4), the amount of packets delivered to a clone node is distributed unevenly by a simple set number of clone nodes; it changes according to the topology and variables based on the user mobility path. A seamless live video streaming service is made possible in a high-mobility vehicle environment using the packet distribution algorithm.

The distributed content packets are forwarded to the selected clone node through the calculated path, which is composed of the selected relay nodes based on the CNQI. Each selected relay node has a time table for forwarding the distributed content packet, in order for the the packets to get distributed. Even if the user vehicle has high mobility, the selected relay nodes and the clone nodes can form a link connection based on the time table of the nodes and the trajectory of the vehicle. The method to obtain a seamless link connection between the user vehicle and the clone nodes is explained in the next Section 3.5.

### 3.5. Packet Delivery Synchronization

After the content packets are forwarded to each clone node, the user vehicle moving toward its destination based on its trajectory receives the content packets from the clone nodes when it passes through the connection area of the clone nodes. In urban road-map distinct features, vehicles naturally experience traffic interruption due to signal lights, traffic, and other factors.

Figure 2 shows the packet delivery synchronization from a clone node to a user vehicle based on the packet calibration approximately. For a clone node, which has an extensive connection coverage, it is challenging to maintain the connection with the user vehicle until the allocated packets are forwarded. There are three main challenges, because the clone node is an end node toward the user vehicle that is streaming data about the destination. First, a clone node must maintain a connection with the user vehicle to achieve QoS and QoE for the user. For the QoS and QoE satisfaction of a user, we must ensure a seamless connection with the user vehicle with a low loss rate and short delay time to improve the service. Second, we have to estimate the data distribution for each clone node by data partitioning because there are two or more clone nodes in the trajectory of the user vehicle. Third, the clone nodes exchange the streaming data owing to real map scenarios, such as traffic, signals, and obstacles. The proposed scheme has a synchronization phase to overcome these issues. The synchronization phase consists of four steps to maintain the connection with the user vehicle: message exchange for recognizing the user vehicle, delay calculation for packet loss reduction, mobility estimation for seamless connection with the user vehicle, and packet calibration for optimization of the synchronization phase. These are performed progressively in order of time.

The message exchange task is the first step of the synchronization phase. In the topology, there are various vehicles on the road within the communication range of a clone node. We have to identify the right vehicle to transmit the streaming data to the user. To satisfy the conditions, when a random vehicle enters the clone node communication range, the clone node exchanges an ID qualifying message with the vehicle to determine if the vehicle is the target. The basis is the data packet header with the ID of the data-requested vehicle.

After the massage exchange, we calculate the data forwarding delay in the user vehicle to support the mobility of the vehicle based on the query and reply packets, which is one signal time. The existing protocols do not support the mobility of a vehicle in live streaming services [29,30]. Finally, this leads to high streaming data packet loss, which affects the QoS and the QoE critically. We compute the location of the predicted vehicle after one signal time using the computed delay time from the mobility information of the vehicle. The proposed scheme reduces the packet forwarding delay and supports the mobility of the vehicle with less energy consumption, owing to the estimation based on one signal time. In the topology, many vehicles have link connections with one clone node. Therefore, the bandwidth and power of a node may have a functional overload if the topology does not consider the capability of the nodes and the traffic. When a user vehicle subscribes to the clone node communication area, the clone node confirms the ID of the vehicle and matches the content data. Simultaneously, if the ID matches with the content data, the vehicle forwards its trajectory information to the clone node for computing the delay and the mobility of the vehicle for the transmission of the content data.

Packet calibration is conducted for a seamless link connection with the user vehicle after the next step of the data forwarding delay calculation. Packet calibration considers two scenarios: when the user vehicle is slower and faster than the states based on the estimated time owing to the urban map features, respectively. The allocated content data, $P_{k}$, must be forwarded within the time that the user vehicle has a connection with the clone node until the user vehicle enters the communication range of the next clone node, $Clone_{k+1}$. Concurrently, the user vehicle has to receive the guaranteed content data with a seamless transmission status. Equation (Equation 5) expresses the condition to satisfy the QoS and QoE at the end node.
(5)$$\begin{array}{cc} C_{k} & =Vt_{k}\times \left(TS_{k}\times \beta_{k}\right)\\ \; & =\frac{TD_{k}}{V_{user}\times \alpha_{k}}\times \left(TS_{k}\times \beta_{k}\right) \end{array}$$
where $C_{k}$ is the calibrated data size and $Vt_{k}$ is the time the user vehicle stays in $Clone_{k}$ in real-time. $Vt_{k}$ comprises $V_{user}$, which denotes the user vehicle, and $\alpha_{k}$ denotes a real-time traffic factor based on the traffic rate with the saturation flow rate [Flow rate] in the communication range of $Clone_{k}$. $TD_{k}$ denotes the trajectory distance of the user vehicle in the communication range of $Clone_{k}$. $TS_{k}$ is the transmission speed of $Clone_{k}$, and $\beta_{k}$ is the bandwidth factor based on the usage of capacity of $Clone_{k}$

We consider two conditions for seamless transmission with high QoS and QoE. Condition $P_{k}>C_{k}$ corresponds to the case in which the user vehicle is unavailable to obtain all content data owing to inadequate communication time or disturbance in communication speed. Under this condition, $Clone_{k}$ forwards the remaining content data to $Clone_{k+1}$. Condition $P_{k}<C_{k}$ corresponds to the case in which the user vehicle has more time for transmission than the computed time in the packet distribution phase, owing to the traffic in the range of $Clone_{k}$. In this case, $Clone_{k}$ computes and requests the requisite content data from $Clone_{k+1}$. The request content data size depends on the content playtime of the users. We formulate the following expression for $R_{k}$, which is the request requisite content data that has been distributed to $Clone_{k+1}$:(6)$$R_{k}=(|est_{k}-Ct_{k}\left|\right)\times p_{bit}+Guardband$$
where $est_{k}$ denotes the estimated time that the user vehicle stays in the communication range of $Clone_{k}$, and it is computed in the packet distribution phase. $Ct_{k}$ denotes the calibrated time at $Clone_{k}$. $p_{bit}$ denotes the playing bit rate of the content video data. We set guardband as in the packet distribution optimization phase. It maintains a seamless video streaming service and prevents the decrease in the QoS and QoE.

Figure 3 shows the flow chart of the Clone streaming service and a brief summary of the proposed protocol. As we mentioned above, the proposed protocol supports a seamless video streaming service and high mobility with stable QoS and QoE. The following section will talk about the simulation experiments that prove the better performance of the proposed protocol compared with the other previous works we mentioned in early this paper.

## 4. Performance Evaluation

In this section, we present the evaluation of the performance of the proposed scheme called Clone by comparing with the existing protocols. For this, we implemented both the existing protocols and Clone in the NS-3 simulator. In the next subsections, we first explain the simple basic concepts of the existing protocols for implementing them in the simulator, and subsequently describe the simulation model and scenario. Finally, we compare the performance of Clone with those of the existing protocols in diverse aspects based on the simulation results.

### 4.1. Comparison Protocols

For evaluating the performance of Clone, we use four existing protocols as comparison protocols. The first one is BAHG [21], which provides a basic non-streaming service based on the usage of backbone vehicles and a greedy forwarding in position-based routing. The second one is TSF [23], which provides a basic non-streaming service using the trajectory information of a destination vehicle. The third one is DQLTV [29], which provides multimedia streaming services by supporting only the QoE to a destination vehicle. The fourth one is GeoQoE [30], which provides multimedia streaming services by supporting both QoS and QoE to a destination vehicle.

The Clone scheme is compared to four other schemes: a basic non-streaming service for the TSF scheme, mobility prediction scheme for the BAHG scheme, streaming service for the DQLTV scheme, and QoE-based streaming service scheme. The comparisons are conducted to show the advantages of our clone-based proposed scheme in high-packet live video streaming service with mobility support as well as the performance improvements over the other categorized schemes. Video streaming through VANETs is challenging, primarily owing to the amount of data forwarding. A high-definition quality video streaming service under the Internet scheme has a required maximum flow of 6 Mbps. Thus, with the WLAN network using the 802.11p standard, the binary throughput is approximately 54 Mbps for providing QoS and QoE over VANETs. However, using an ideal environment for obtaining the best QoS and QoE performance in real VANETs is unrealistic. Consequently, we focus on real map data for location setting of the RSUs and relay nodes for the simulation.

Regarding basic non-streaming service, TSF is the representative scheme using vehicle trajectory for comparison to the proposed scheme. In TSF, a localization system is installed in vehicles to provide the movement path of a vehicle moving toward the destination. Using the mobility information from the localization system in the TSF scheme, TSF calculates the expected delay of a road segment for video packet forwarding. For basic streaming service, DQLTV is the representative protocol for comparison to the proposed scheme. DQLTV utilizes multi-hop level forwarding and delay, quality of the link, and a link lifetime aware routing protocol suitable for video streaming service. DQLTV adopts the role of maximizing the link quality and the link lifetime as well as minimizing the distance between the hops and the overall delay for reducing the frame loss. However, there are unconsidered factors that affect the overall performance of the video transmission, such as QoS and QoE. The Geo-QoE scheme is an improved scheme using QoS and QoE factors to enhance the QoE of a user vehicle. For the mobility support scheme, BAHG is the representative protocol for comparison of the mobility support part for video streaming services. BAHG derives the trajectory of a vehicle for minimizing the number of intermediate intersection nodes while considering connectivity. Moreover, backbone nodes track the movement of a source and a destination vehicle for providing the connectivity status around an intersection. We consider three of the challenges in a video streaming service in the experiments. Our proposed scheme is compared against multiple QoS and QoE factors, respectively.

### 4.2. Simulation Model

#### 4.2.1. 802.11p Standard

In VANETs with video streaming services, wireless communication systems encounter challenges due to the characteristic of the wireless environment. First, a wireless communication cannot provide the same satisfaction level to all users in terms of the QoS and the QoE, and it cannot cover extensive content in a vehicle, such as infotainment, which is composed of massive data content. Moreover, the massive data content is associated with a risk of packet duplication, forwarding error, and other problems that decline the user satisfaction. Second, mobility changes of the vehicles in VANETs are inevitable, and a wireless communication channel can change rapidly owing to the variation in the distance between the source and the user vehicle, which have high mobility. Interference is caused by mobility changes, and urban map characteristics are distinct consequences of the mobility of a vehicle. Third, the video streaming service can be conducted on a support, such as 802.11p. Furthermore, delivering each chunk to the relay and clone nodes in real time is susceptible if the protocol is derived from other standards.

However, the 802.11p norm is the third standard using waveband 5.835 GHz to 5.923 GHz frequency band with orthogonal frequency division multiplexing (OFDM) signaling. The communication protocol used in the existing ITS environment is DSRC, mainly used for short-distance communication. However, with the ever-changing Internet of Things environment and the development of smart cars, it was necessary to develop a communication technology that can recognize traffic conditions on the road in real-time to prevent traffic accidents in advance and smoothly adjust the traffic flow. Wireless Access in Vehicular Environment (WAVE) is a standard developed to meet these requirements. WAVE is a wireless communication technology that delivers dangerous situations on roads or vehicles to vehicles in a vehicle network with high-speed mobility (up to 200 km/h) and short communication exchange time. V2X (V2V, V2I, V2N, V2P) communication is possible in real-time, even in high-speed driving conditions. It is a source technology that prevents accidents by processing real-time information such as road conditions and vehicle accidents.

#### 4.2.2. Video Contents Structure

The video content structure used for the experiments, H264 AVC flux, is composed of image frames. H264 AVC is a global video structure also known as MPGE-4 part 10, and it is a successor of earlier standards MPEG-2 and MPEG-4. The simulation model used H264 AVC for video streaming scenarios to forward the content packets. H264 AVC has variable bit rates depending on the color of the composed image in the video. It is suitable for comparing with various color bind videos based on QoS and QoE factors.

#### 4.2.3. QoS Parameters

QoS is a video content quality measurement based on user experiences. QoS parameters can be categorized as opportunities to provide high-performance video streaming services. To maximize the QoS that a user experiences, several parameters—bandwidth media, control of jitter and controlled period, and decreased packet loss—must be considered and tuned. In our simulations, end-to-end delay and packet delivery ratios based on the jitter as well as the packet loss that results when forwarding the video content data were considered as QoS parameters. As already mentioned about the QoE, if video streaming does not consider the QoS in packet forwarding, the protocol will encounter crucial problems in the transfer and routing process of voice and video quality.

#### 4.2.4. QoE Parameters

The perceived video quality is measured by the two most commonly used QoE parameters: MOS and peak signal-to-noise ratio (PSNR). The PSNR is the standard metric for measuring the objective video quality. This parameter is expressed as a function of the mean standard error between the original and received video frames. If a video frame experiences either transmission or overdue loss, it is considered to be dropped and may be concealed by copying the payload from the last received frame before it. Metric SSIM used as an image/video metric to measure the received frame quality based on its structural, luminance, and contrast similarity. The SSIM, measured using MSU tools, improves the MOS and PSNR metrics by revealing the perceived quality of the received video sequences. The SSIM differs from the PSNR because it approximates structural distortion, instead of the pixel-by-pixel errors, to evaluate useful information to human eyes.

### 4.3. Simulation Scenario

In our experiments, the 802.11p standard for ad-hoc network QoS is derived. Three classes of video files with variable bit rates of 30 fps are considered. Each video file is composed of night view scenes (which combine multiple colors), ocean scenes (combining blue colors), and sunset scenes (combining red colors). The resolutions of the three videos are 720P high definition (HD), 1280 × 720 progressive scan, and 923,600 pixels (0.92 MP) per frame. The routing protocol for the packet transmission to the relay and clone nodes uses GPCR and AODV. We consider two binary flows: 11 Mbps as the minimum speed and 54 Mbps as the maximum speed. The detail of the parameters are in Table 2 below.

We use a network simulator (NS-3) to simulate the video streaming service with high packet data on the Seattle city map. The vehicles requesting messages and receiving the video content are placed on the Seattle city map of the network. A set of 50 requesting vehicles is placed in an area of 3000 m × 3000 m, and the 300 nodes that take the roles of relay and clone nodes located in the network area follow the features of the urban map. The mobility model of the vehicles on the map follow random mobility with the Manhattan mobility model. The CNQI information is already computed and shared with the neighbor nodes when a scenario is initiated, such as a real-map scenario, for the proposed scheme experiments. Therefore, we ignore the delay of the topology setting time for the overall delay estimation time.

### 4.4. Simulation Results

Figure 4 shows the End-to-End overall delay on each compared protocol with basic scenarios for video streaming services to forward the video contents data. The basic scenario is that the source forwards the video files with boundary conditions in 11 Mbps to 54 Mbps depending on the protocols algorithm to the user vehicle, which has the mobility. Figure 4 proofs TSF and BAHG are not suitable to give QoS satisfaction to the users in live video streaming service circumstances due to the two protocols only using a single packet transmission scheme to forward the data. The single-packet transmission scheme causes various problems in the video packet transmission process towards the user vehicle because of the mentioned problem. In the streaming service, the End-to-End delay is one of the keys to satisfying the user QoS due to the streaming video potentially being paused to buffer if the connection quality is not enough to catch up with the active playtime. Therefore, protocols must support the declining delay and the loss rate in video data transmission. TSF and BAHG only have trajectory data to predict and calculate the user vehicle’s destination with the route. There is nothing to support the mobility of the vehicles in real-time. However, Geo-QoE, DQLTV, and Clone have a supportive way to decline the delay by monitoring the vehicle’s quality of the links and qualities, links lifetime, the hop-count, and cumulative distance. Moreover, Clone has a data caching mechanism on the clone nodes not to get frame losses due to the mobility of the vehicles.

Figure 5 shows the packet delivery ratio during the simulation time in the basic scenario. We can see the gap between the two groups of the protocols. TSF and BAHG do not support video file data forwarding well due to a lack of the link correction algorithm, even if the two protocols have the trajectory information to support the vehicle’s mobility. In a single packet or small data transmission, link disconnection with the user vehicle is not a critical issue in the delivery ratio. The two protocols have alternative paths to forward the small data. However, suppose the file goes to mass data, even if a single packet gets losses. In that case, intermediate nodes in the two protocols take overloads due to the operation and computation quantity for the losses data and moving vehicles. Otherwise, DQLTV, Geo-QoE, and Clone consider the link quality of the nodes in the topology to forward the video files packet with less loss rate. Geo-QoE and DQLTV have the next forwarding vehicle’s selection scenarios using the collected vehicle’s data of network quality information. Additionally, Clone has the synchronization phase for the link connection between the user vehicle and the clone nodes with the distributed content data. Consequently, the frame loss has declined, and the packet delivery ratio has increased more than other protocols due to the QoS and QoE supports in the proposed scheme.

Figure 6 shows the packet delivery ratio for node hops to the destination. In this simulation, we assume that the user vehicle has a static location to see how the protocols derive the utilization of information in the topology uniformly. TSF and BAHG only consider the trajectory of the vehicle with the intersection node density on the road. It leads to an imbalance of energy and bandwidth consumption in the whole topology due to the few specific nodes located in the comprehensive communication place that have derived frequently. It becomes a more critical problem when the user vehicle gets mobility due to the computation quantity on the nodes. However, DQLTV, Geo-QoE, and Clone have a way of identifying the link quality of the nodes. It attests to the traffic distribution in the topology interactively. Furthermore, because the Clone uses CNQI information and the information-sharing system before finding the forwarding path to the clone nodes using the relay nodes, the overall traffic distribution shows better performance than other protocols.

Figure 7 shows the average delivery ratio according to the average speed of the vehicle. In the simulation scenario, the vehicle has random mobility combined with Manhattan mobility. Each user vehicle has different speeds on the road, and we did experiments with two conditions: 11 Mbps as the minimum speed and 54 Mbps as the maximum speed. Figure 7 is an average graph with the two conditions’ results. When the vehicle’s speed goes to high-speed status, respectively, the computing process of the source node gets increased due to the change of the forwarding path to the user vehicle. Consequently, the topology changes have occurred continuously due to the vehicle’s mobility. Clone uses the user vehicle’s trajectory to predict the user vehicle’s mobility using Clone node selection and synchronization processes. The clone node is placed on the user vehicle’s trajectory based on the user vehicle information and distributed content data size. Hence, the received distributed content data in the clone node is forwarded to the user vehicle. Moreover, four steps of the link synchronization algorithm prevent link disconnection between the user vehicle and the clone nodes for QoS and QoE. Furthermore, the packet distribution algorithm is flexible in responding to the vehicle’s mobility in the topology. However, TSF and BAHG only use the trajectory of the vehicle to forward the packet to the predicted place. The two protocols have no process to support the video streaming content data due to the two protocols working for the single packet transmission, DQLTV and Geo-QoE cannot support the vehicle’ mobility in real-time with an adaptive method which the proposed scheme has. The graph shows the distinction between Clone and other protocols in the delivery ratio.

Figure 8 shows the average delay of the overall process in the protocols depending on the average speed of the vehicle in the two binaries. In the video streaming service, QoS and QoE satisfaction of the user is the most emphasized factor. To carry out the result for the factor, we need to minimize the delay in the data forwarding processes of the protocols. TSF and BAHG research the user vehicle at every moment when the vehicle changes the link location due to mobility. The forwarding processes of the two protocols execute researching messages, forwarding several times until they find the user vehicle in the communication range without any obstacle. The tasks waste time and affect the QoS and QoE in the video streaming service. DQLTV uses the neighbors’ table to forward the content data to the user vehicle. It is quite useful in monotonous forwarding circumstances with small content data forwarding. However, the table with RSU to transmit the extensive contents data cannot deal with the high mobility circumstances when the vehicles move fast through the many intersections. Clone derives reliable relay nodes with the clone nodes, which has a high-level link connection power computed by CNQI. The idea of Clone to figure out the nodes’ condition information, can handle the changing topology due to the mobility of the vehicle. The graph verifies that Clone has comprehensive methods to deal with the mobility of the vehicle. Consequently, TSF and BAHG are not suitable to support the live tracking vehicle with a high mobility situation depending on the vehicle speed. There is no process to predict the mobility of the vehicle in dynamic changes. Although GeoQoE and DQLTV have the process of supporting the vehicle’s high mobility circumstances, the mentioned two protocols and Clone have a different process to calculate the topology status. Moreover, Clone has the synchronization phases and the packet distribution algorithm to optimize the relay node data forwarding and selection. Therefore, the defective part of the GeoQoE and DQLTV does not affect Clone due to the process we mentioned. We demonstrated that Clone could give a better streaming service than other comparison protocols in terms of QoS. The following figures show experiments of QoE factors with the three video files mentioned at the beginning of the simulation parts.

Figure 9 shows the PSNR (Peak Signal to Noise Ratio) for each total image frame. We use the three different image files mentioned above. The video file for QoE, the simulation, has been divided into three equal parts in the experiments: sunset, night view, and ocean. The values of PSNR of the tested videos are decreasing on TSF, BAHG, Geo-QoE, and DQLTV streaming protocols when the content’s data frames get increased because the protocols cannot support the live movement tracking mechanism Clone has. Clone supports the content data caching process on the relay nodes and clone nodes to forward the distributed content for the user vehicle, which has high mobility in real-time. Due to this reason, the values hold a stable variation with Clone.

Figure 10 shows the CNQI score for the AP density and communication range in the proposed scheme. The communication range of AP and the density of the vehicle affect the performance of the proposed scheme. The CNQI score is one of the main variation factors to show the overall performance of the proposed scheme. In the experiment, the communication range bound is 200–800 m due to the 802.11p WAVE standardization. The AP density is determined by 300 AP numbers basis. Moreover, random mobility vehicles moving on the roads support the transmission as intermediate nodes. With the AP numbers and the vehicles, we calculate the AP density rate for the experiment results. As the AP density rate increases, the number of AP and vehicles participating in the video streaming transmission increases. Above a 60 density rate from the result, the proposed scheme has enough communication APs and vehicles. Therefore, the increment of the CNQI score has decreased.

Figure 11 shows the Packet Delivery Ratio (PDR) for AP density and the communication range in the proposed scheme. PDR is affected by various factors in the proposed scheme, such as vehicle density, distribution of APs, communication condition of APs, and others. The result shows that the PDR performance depends on the critical factors mentioned before in the proposed scheme. When the communication range has increased, the number of communication capable APs and the vehicles increases if there is no limitation to the density rate. However, we set the limitation of the density rate from 30% to 70% for a more realistic scenario. If the communication range increases with a fixed density rate, the distance among the APs and the vehicles for the communication changes sparsely. TSF and BAHG depend on the neighbor nodes and APs in a one hop range. Due to the simulation scenario, communication failure may occur continuously if the node density changes sparsely. GeoQoE and DQLTV were also affected by the node density and the distance between the neighbor node because of the differentiation of the topology information analysis algorithm. Thus, the probability of packet loss in the transmission is increased. The result shows the decrement of PDR more significantly in the low-density rate due to the sparse density.

However, Figure 12 shows that the Clone scheme shows better APDR performance than the comparison protocols with the same experiment condition in Figure 12. Due to there being no support for the video packet transmission in TSF and BAHG, the result shows a low Average Packet Delivery Ratio (APDR) in overall performance. TSF and BAHG have to re-transmit the lost packet when the vehicle is moving without any backup plan. Therefore, with the high mobility situation and video packet transmission, TSF and BAHG have a low-APDR even though they use the trajectory information of the vehicles for the packet transmission. Otherwise, DQLTV and GeoQoE show better APDR than TSF and BAHG because DQLTV and GeoQoE have video packet transmission processes and QoS and QoE support applications. However, DQLTV and GeoQoE do not have a real-time support application for live streaming services. Clone scheme has a clone node mechanism and a caching mechanism to support real-time user vehicle tracking in live video streaming. Thus, the result proves that the Clone scheme shows better APDR performance with the various density rates and the communication range conditions.

Figure 13 shows the average delay for communication range and density rate in the proposed scheme. As we mentioned above, various communication range conditions and density rates affect the overall performance of the proposed scheme. The result shows that the average delay gets increased depending on the increment of communication range and the decrement of the density rate. Those two factors affect the distance among the relay vehicles and communication APs to the destination. In addition, the APDR of the proposed scheme also affects the average delay in transmission due to transmission failure from real-time mobility support. Thus, the communication delay gets increased.

Using the same experimental conditions in Figure 13 and Figure 14, they show the average delay for the communication range and the average density rate. The average delay gets affected by communication distance and PDR due to transmission failure. TSF and BAHG, with a single packet forwarding mechanism based on trajectory information, show a higher average delay than other comparison protocols, including the proposed scheme. The reasons for the two trajectory-based protocol results could be mentioned as low APDR, low efficiency of packet failure handling, and absence of video streaming transmission process. Otherwise, DQLTV and GeoQoE show better performance in average delay because those two protocols are based on video streaming transmission considering either the QoE factor or the geographic information combined with QoS QoE. However, DQLTV and GeoQoE are not suitable for real-time video streaming transmission. There are no real-time mobility tracking mechanisms to support the mobility of the user vehicle in the two protocols. The proposed scheme has four phases for the caching mechanism with the Clone technique to support the real-time video streaming service. Thus, the proposed scheme can reduce the probability of packet failure from mobility support. Therefore, this leads to a low average delay compared to other comparison protocols.

Figure 15 shows PDR for the number of video streams. We used three types of image video files, Sunset, Night view, and Ocean. Using the mixed video file, which included the three types of video, we experimented with the influence of PDR depending on the video file size. Increasing the number of video streams means the increasing the increment of users in one AP. In other words, the usage of the bandwidth of the AP gets increased. Consequently, the increment of the bandwidth of the AP affects the performance of the AP. The PDR results of TSF and BAHG are worse than other streaming service protocols because the two protocols are not suitable for video streaming transmission. Th refore, there are no noticeable changes while the number of streams is increasing. However, the proposed scheme shows noticeable differences between the other two streaming protocols. DQLTV and GeoQoE have either mathematical algorithms or use geographic information to select the next-hop candidates. However, there is no live status applying technique for APs in the topology. Therefore, as time goes by, the two protocols cannot update the changes of the overhead of APs. The proposed scheme has the CNQI calculation and status update process for real-time support. Thus, the proposed scheme can use the resource of APs in the topology uniformly, no matter the time that passes. The result shows a better PDR than other comparison protocols in the experimental conditions.

Figure 16 shows that the MOS (Mean Opinion Score) in the moving vehicle depends on the image files. MOS values have been graded based on user experienced quality. MOS values of the protocols in the experience have measured the comparison between the original file and the forwarded video files using “MSU Video Quality Measurement Tool”. As the other experiments already showed, the protocols’ QoS affect user QoE directly due to the protocol’s forwarding abilities and the mobility supportability. The Clone scheme has QoS and QoE supporting techniques, which give a solution for tracking a moving vehicle with less image distortion. No matter which image structure the protocols forward, Clone shows the better performance in the MOS value of moving vehicle.

Figure 17 shows SSIM values in mobility support by the protocols. SSIM results show the similarity of visual image quality based on the user experience with human eyes. Moreover, SSIM does not consider a numeric error between the original video file and the transferred video file. The graph shows ghat Clone gives better video transmission quality than other protocols in the overall images. It means that Clone transfers the original video file with less image distortion in QoS and QoE. The differences in the protocols’ supporting the process lead to the quality of the video. The measured values of SSIM and MOS prove Clone gives better QoE due to keeping a stable packet delivery ratio variation in the overall process. It comes from the results of the QoS factors experiments.

## 5. Conclusions

In this article, we have proposed live video streaming services that support the mobility of the vehicle by the Clone scheme. We have formulated CNQI to figure out the topology and status of the nodes in video transmission to select the best relay nodes, a packet distribution optimization method with guard band for mobility support, and a link connection phase for a seamless video streaming service that reacts to various unforeseeable circumstances. Based on the CNQI algorithm, we developed high QoS and QoE video streaming services with high mobility situations in real-time.

The proposed protocol, called Clone, derives a conclusion from the various processes in this paper to solve the three challenging issues in the video streaming service of VANETs. In large video streaming circumstances with high mobility, such as in moving vehicles, the video data should be divided into many chunks. The divided chunks have to be forwarded to one of the nodes on the requested vehicle trajectory. The Clone protocol has a process to distribute the requested video content data forward to the clone APs. The data distribution algorithm from the proposed protocol gives enormous advantages in load balance in the topology and the seamless data transmission. The proposed protocol considers the overall topology information to deliver the distributed video content data to Clone APs. It prevents intermittent data loss while the relay node is delivering the requested video data. This leads to a stable transmission of the proposed protocol.

Moreover, the proposed protocol calculates the topology QoS and QoE information by CNQI measurement. The load balancing of the topology in the video streaming service shows better performance from the proposed protocol than other topology measurement protocols in the simulation results. Various factors that do not get the eye in the compared protocols, such as the energy of the APs, the number of subscribers, the bandwidth of the AP, and others from the CNQI measurement factors, accepted the proposed protocol to analyze the topology information. To improve the QoS and QoE in a video streaming service with a high mobility situation, the CNQI measurement in the proposed protocol gives better results than other compared protocols in QoS and QoE. As the proposed protocol distributes the contents data to forward the video packets, the data caching process should be stored until the requested vehicle receives the data. The compared protocols do not have an effective caching mechanism for QoS and QoE. Hence, the simulation results show that the proposed protocol is adaptable in various topology changes such as density, mobility, and energy of APs.

The experiments have been tested under various conditions such as communication range changes, video file sizes, hop counts, speed of the vehicles, and AP density in the topology. Thus, we draw simulation results about the end-to-end delay, PDR, PSNR, MOS, and SSIM. The results show that the Clone scheme gives 10% to 20% better results than other video streaming services in overall performance with QoS and QoE and 60% to 80% better results than other comparison packet delivery routings. The simulation results have verified better QoS and QoE in the massive mobility changes of urban map features. Besides, the results have demonstrated that our proposed scheme could perform much better than other schemes in QoS and QoE with load balancing.

## Figures and Tables

**Figure 1 sensors-21-07368-f001:**
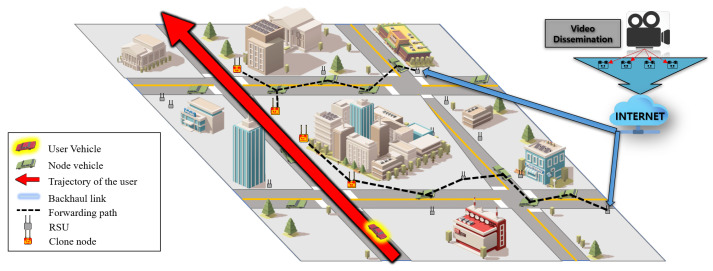
Clone streaming service overview.

**Figure 2 sensors-21-07368-f002:**
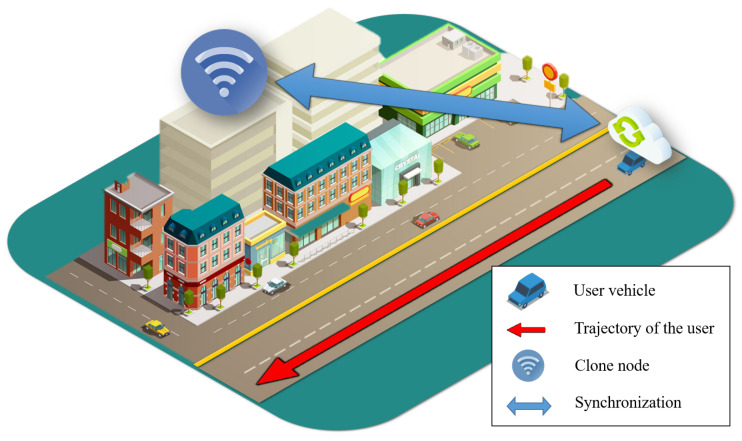
The packet delivery synchronization from a clone node to a user vehicle based on the packet calibration.

**Figure 3 sensors-21-07368-f003:**
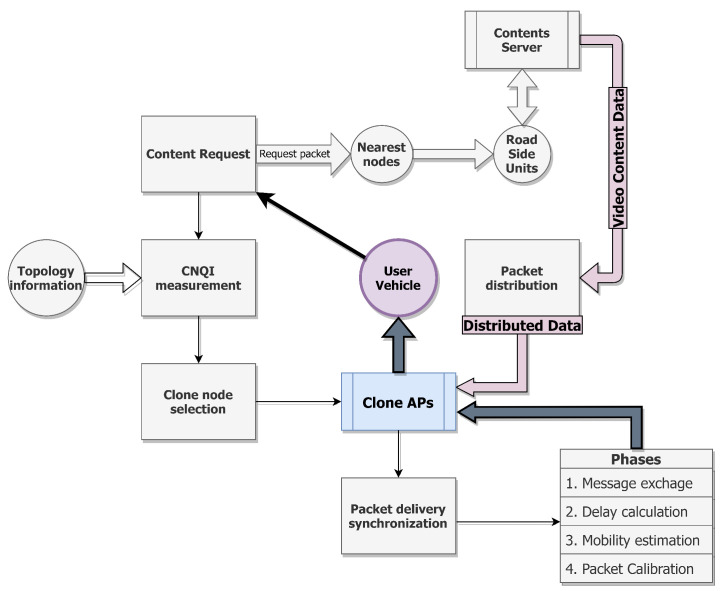
Flow chart of the Clone streaming service.

**Figure 4 sensors-21-07368-f004:**
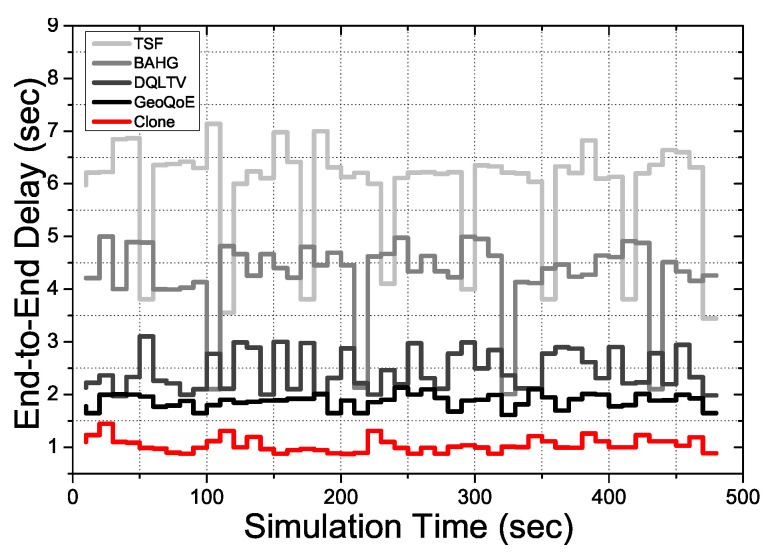
End-to-end delay results of the overall simulation time.

**Figure 5 sensors-21-07368-f005:**
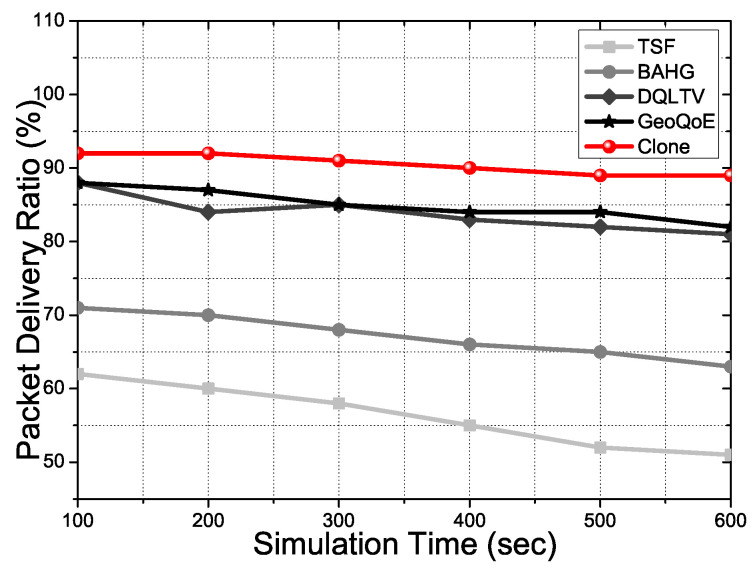
Packet delivery ratio of the overall simulation time.

**Figure 6 sensors-21-07368-f006:**
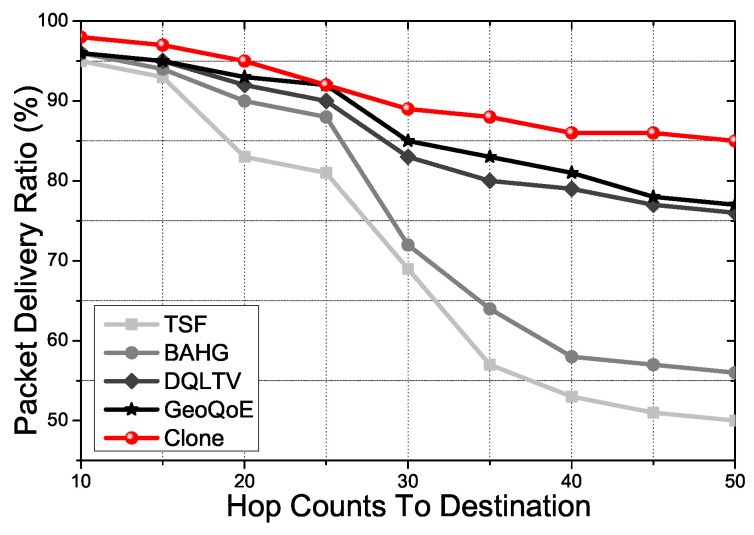
Packet delivery ratio for hop counts to the destination.

**Figure 7 sensors-21-07368-f007:**
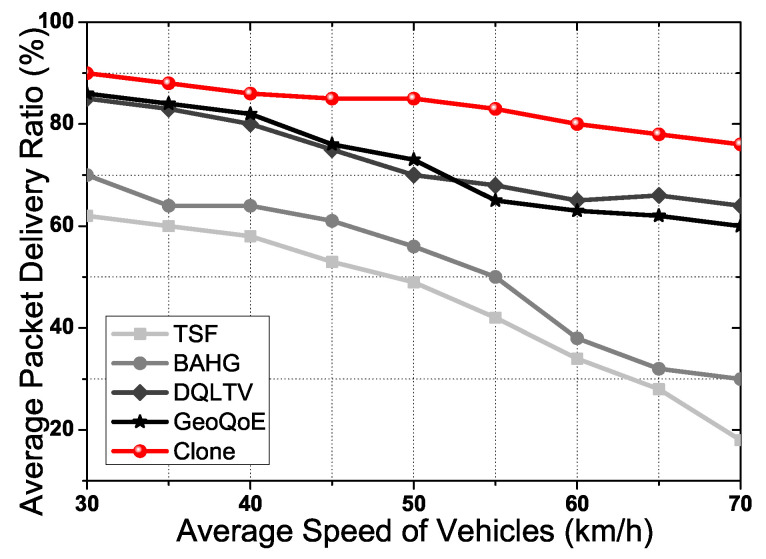
Average packet delivery ratio for average speed of vehicles.

**Figure 8 sensors-21-07368-f008:**
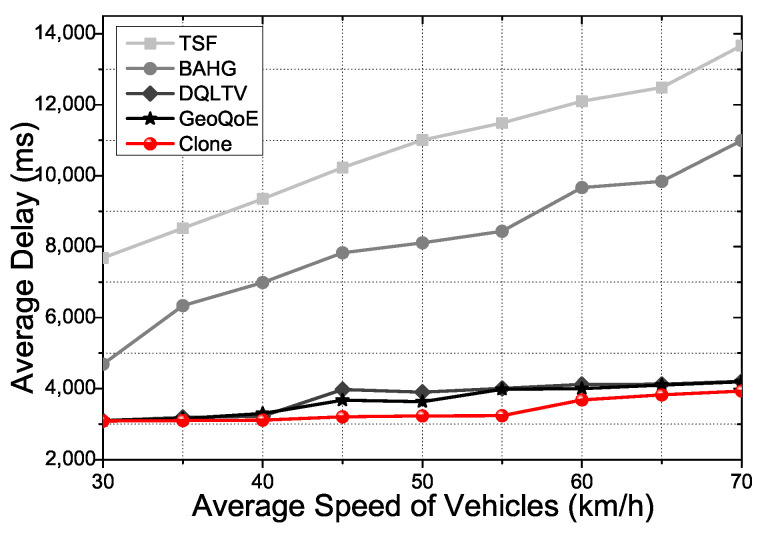
Average delay for average speed of vehicles.

**Figure 9 sensors-21-07368-f009:**
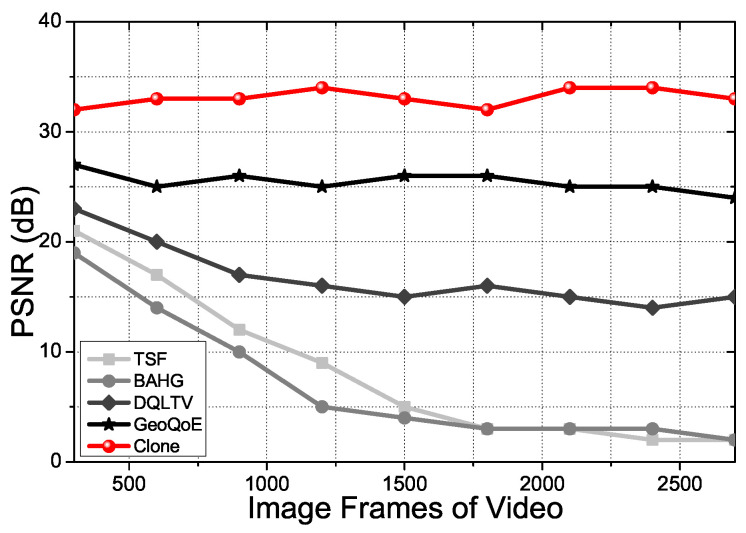
PSNR for the image frames of video.

**Figure 10 sensors-21-07368-f010:**
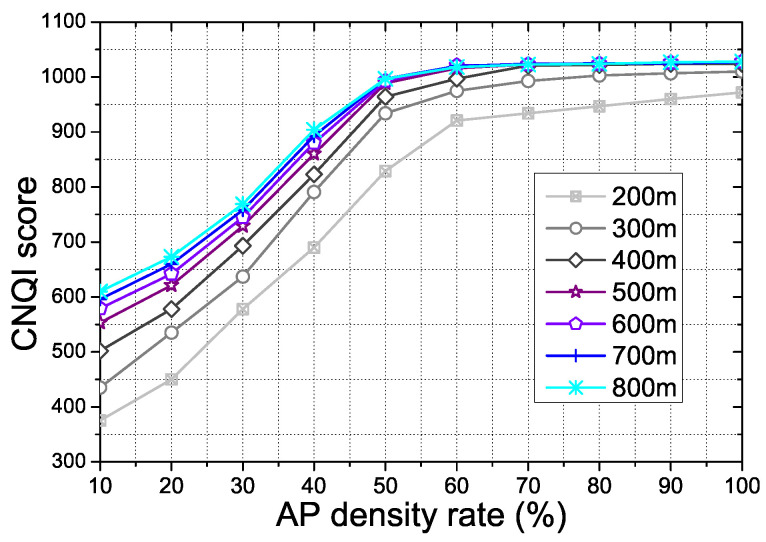
CNQI score for the AP density and communication range.

**Figure 11 sensors-21-07368-f011:**
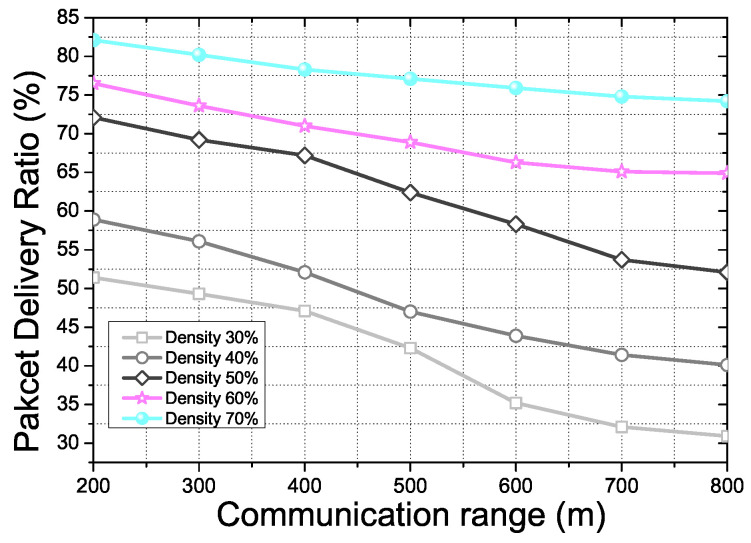
PDR for AP density and communication range.

**Figure 12 sensors-21-07368-f012:**
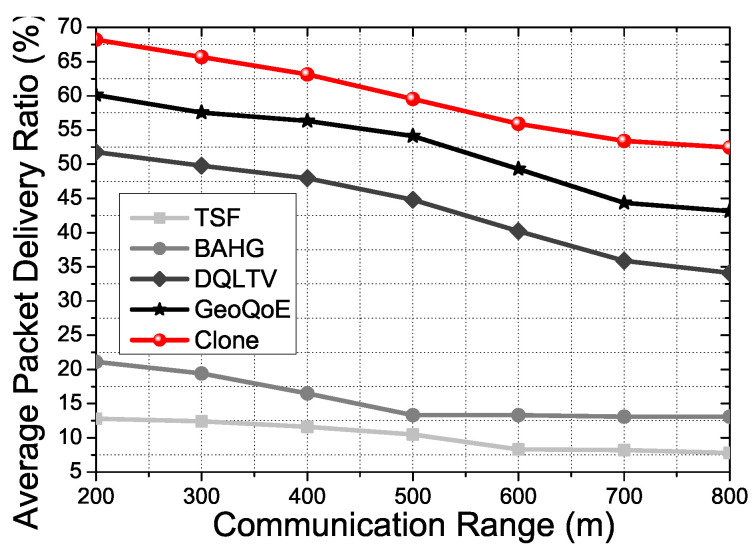
APDR for communication range and average density rate.

**Figure 13 sensors-21-07368-f013:**
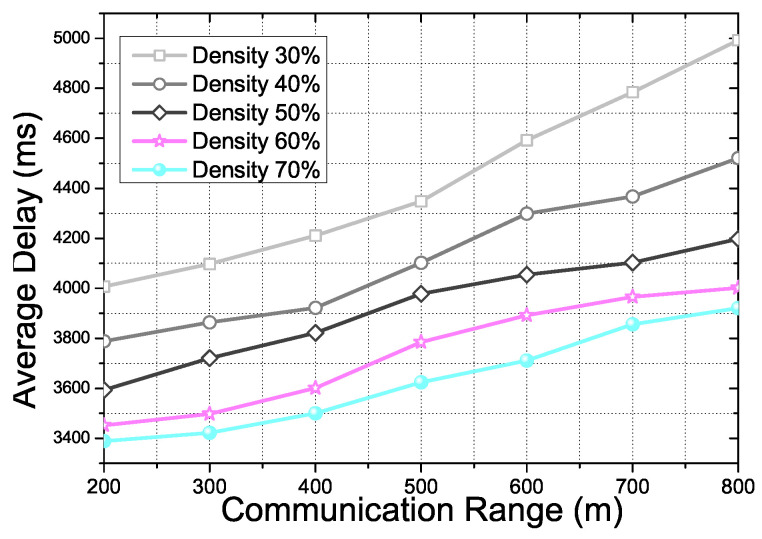
Average delay for communication range and density rate.

**Figure 14 sensors-21-07368-f014:**
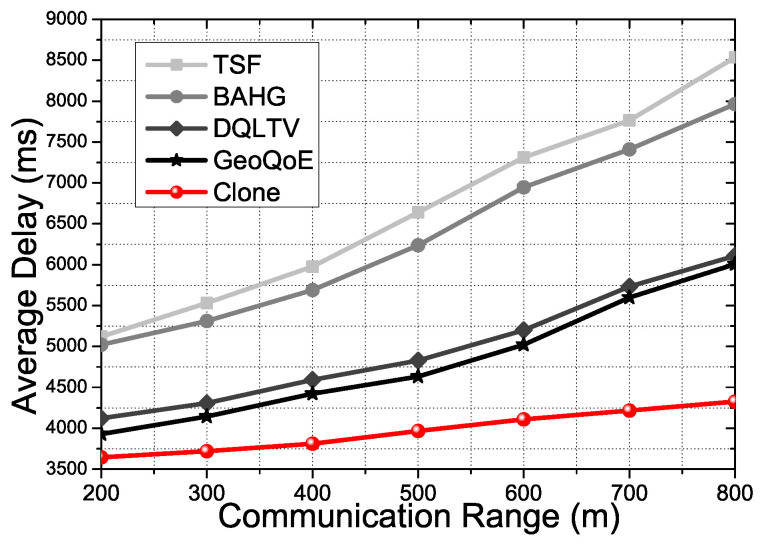
Average delay for communication range and average density rate.

**Figure 15 sensors-21-07368-f015:**
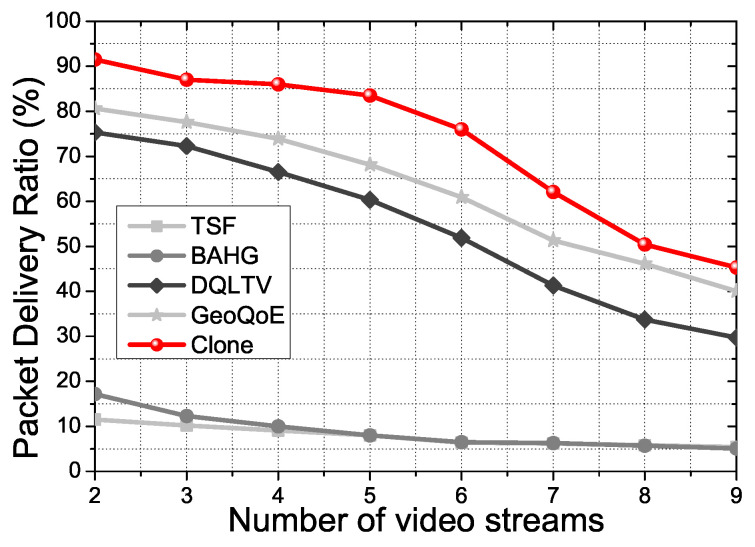
PDR for the number of video streams.

**Figure 16 sensors-21-07368-f016:**
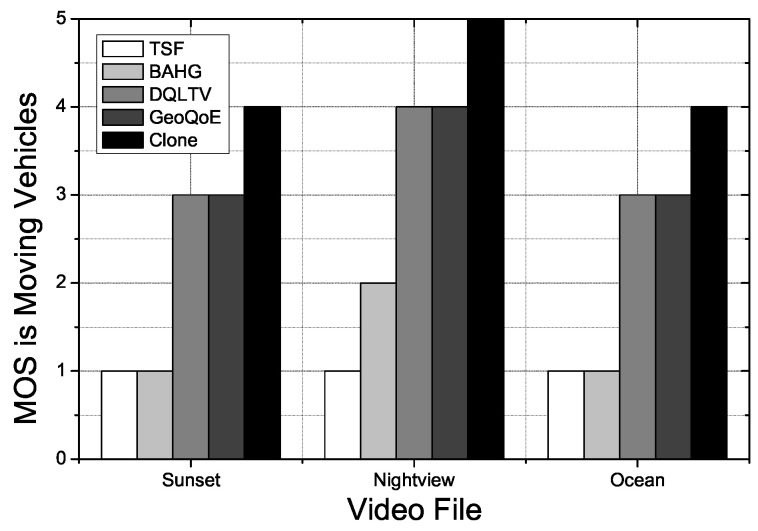
MOS for each video file in a mobility scenario.

**Figure 17 sensors-21-07368-f017:**
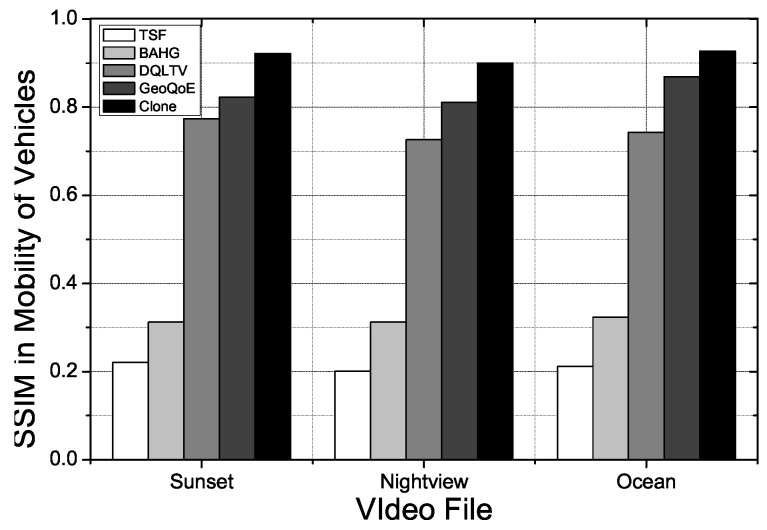
SSIM for each video file in a mobility scenario.

**Table 1 sensors-21-07368-t001:** The summary of the related works on video streaming services in VANETs.

Ref. No	Features	QoS & QoE	Video Streaming	Mobility	Real-Time	Data Cache	Approach
[16]	MANET, Mesh networking, Backpressure routing	X	X	X	X	X	TBR
[17]	Based on the Bellman-Ford algorithm, solving routing loop problem	X	X	X	X	X	TBR
[39]	Optimal Dynamic source routing, support host density and movement rates	X	X	X	X	X	TBR
[18]	Greedy forwarding, Geographic routing	X	X	X	X	X	PBR
[19]	Greedy forwarding, Geographic routing, Junction routing	X	X	X	X	X	PBR
[40]	Combined with position-based routing with topological knowledge	X	X	X	X	X	PBR
[20]	Junction-centric logic and adoption of source routing mechanisms	X	X	O	X	X	PBR
[21]	Hop greedy algorithm, back-bone node concept	X	X	O	X	X	PBR
[22]	Privacy-Preserving trajectory sharing scheme, Link delay model	X	X	O	X	X	TJBR
[23]	Packet and vehicle rendezvous point, vehicle delay distribution and packet distribution.	X	X	O	X	X	TJBR
[24]	Efficient reception point, An arithmetical model based on a road map information	X	X	O	X	X	TJBR
[25]	Multipath multimedia transmission, Enhanced method of GPSR	X	O	X	X	X	VPF
[26]	Multipath forwarding scheme using location information, minimization of route coupling effect	X	O	X	X	X	VS
[41]	Enhanced scheme of LIAITHON, receiver-based scheme, route coupling effect algorithm	X	O	X	X	X	VS
[27]	The video reactive tracking-based unicast protocol, point-to-point video streaming	X	O	X	X	X	VS
[37]	Inter-vehicular communication, cross-layer solution	X	O	X	X	X	VS
[42]	A utility-based maximization problem to find the best delivery strategy	QoS	O	X	X	X	VS
[28]	A cluster-based routing, artificial bee colony technique	QoS	O	X	X	X	VS
[29]	Mathematical model to optimize the QoE scores in video streaming	QoE	O	X	X	X	VS
[30]	Numerical next hop selection based on QoS and QoE factors to enhance the user QoE.	QoE	O	X	X	X	VS
[31]	QoE-driven unequal Error Protection (UEP) scheme	QoE	O	X	X	X	VS
[32]	Analysis of Mac layer behavior and utilization improvements	QoE	O	X	X	X	VS
[43]	Markov Decision Process, optimal chunk selection strategy that maximizes streaming quality	QoE	O	X	X	X	VS
[34]	Sending video reporting messages to an AP in the infrastructure	X	O	X	X	X	VS
[35]	Mobility prediction, adaptive selection algorithm for roadside APs and gateway-vehicles	X	O	O	X	X	MPVS
[36]	Mobility prediction, multi-tier caching mechanism, content request distribution	X	O	O	X	O	MPVS
[44]	Mobility prediction, handling bandwidth prediction errors, adaptation algorithm	X	O	O	X	X	MPVS
[45]	Mobility prediction, adaptive bit rate (ABR) support algorithm	X	O	O	X	X	MPVS
[46]	Mobility prediction, a dual time-scale dynamic cache scheme	QoS & QoE	O	O	X	O	MPVS
Clone	Clone mechanism for real-time video streaming service (the proposed scheme)	QoS & QoE	O	O	O	O	RTVS

**Table 2 sensors-21-07368-t002:** Simulation Parameters.

Parameter	Value
Simulated network field	3000 m × 3000 m
AP numbers	Up to 300 (Random position)
Vehicle numbers	50 to 100 (Random mobility)
Communication range	200 m to 800 m
MAC protocol	802.11p
Mobility model	Manhattan random mobility model
Min/Max bitrate	11 Mbps to 54 Mbps
Video Resolution	1280 × 720 (720p)
Video Frame rate	30 Fps
Video files	Night, Ocean and Sunset
Progressive scan	923,600 pixels
Vehicle speed	Average 60 km/h
Simulation time	600 s

## Data Availability

Not applicable.

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
