# Peer review of "Video Packet Distribution Scheme for Multimedia Streaming Services in VANETs"

_sensors, 2021, doi:10.3390/s21217368_

Round 1

Reviewer 1 Report

Paper shows a VANET multimedia streaming distribution based on cloned caching. Client is cloned by a distributor (not clear if results includes mobile or only fixed relays) and routing is thus forwarded to the clone. There is no active route prediction, but coverage is overlapped to allow smooth handover.  Authors also created a metric "CNQI" that intended to depict real time network quality.

Text is well organized and needs English review (semantics). 

There is no data about the necessary caching on relay nodes. As it seems, routes are going to follow the moving target, but authors missed (or it is not clear) a mobile-mobile scenario. Some other simulation suggestions: compare the coverage overlapping demands (cost), compare data duplication on the network (due extensive caching), and discuss security issues regarding cloning.

Author Response

Point 1: Paper shows a VANET multimedia streaming distribution based on cloned caching. Client is cloned by a distributor (not clear if results includes mobile or only fixed relays) and routing is thus forwarded to the clone. There is no active route prediction, but coverage is overlapped to allow smooth handover.  Authors also created a metric "CNQI" that intended to depict real time network quality.

Text is well organized and needs English review (semantics).

There is no data about the necessary caching on relay nodes. As it seems, routes are going to follow the moving target, but authors missed (or it is not clear) a mobile-mobile scenario. Some other simulation suggestions: compare the coverage overlapping demands (cost), compare data duplication on the network (due extensive caching), and discuss security issues regarding cloning.

Response 1: we really appreciate the reviewer's comment on the improvements of this paper. First, as the reviewer’s comment that the explanation of the mobile to mobile communication scenarios, we miss out the explanation on the topology status of mobility in the overview section. In this paper, the moving AP and nodes are user vehicles only. Other internet accecable APs and nodes are distributed fixed locations on the topology. As the reviewer’s comment, we rewrite the network overview to fix the explanation about the status of APs and nodes in the network overview section. And the following comment for caching of relay nodes, in our proposed protocol, only selected Clone node has a task of data caching. The relay nodes toward the Clone node only support the data transmission, and they do not have another task except data forwarding. As the reviewer's comment, for future work, we will analyze the caching mechanism to the relay nodes to improve the network transmission quality in QoS and QoE later. About the cost simulation, Vehicular Ad-hoc networks apply the concept to the vehicle and RSU. The mentioned nodes and APs do not have a limitation of energy consumption. Also, no matter how the nodes take the cost in the video streaming service, the user QoS and QoE are more critical factors to show the performance. However, as the reviewer’s comment, we have a guideline to improve our work for the future. Thanks for the comments and the improvements for future work. And I really appritiate the reviwer pointed out the security problem. However, if the paper considers the security in data transmission, the paper concept should be changed and it take a longer manuscript. However, we consider it for our future works as well. Thanks for the comments

Reviewer 2 Report

[28-32] Since your manuscript is already very long, consider breaking these three key challenges into a numbered list to help break up the significant wall of text.
[121] The first three pages of the document are just a wall of text, please consider adding subsection headers to divide the content like you do in later sections (i.e. Challenges, Background, Contributions, etc.).
[33-83] This literature review is well written, well organized, and well cited. My only suggestion is again to separate it from the rest of the text with a subsection header. Nice work here nonetheless.
[84-115] You provide a well-written, detailed explanation of your proposed Clone solution. Since you went through the effort of identifying the 3 critical challenges earlier in the introduction, can you add a short paragraph describing how the comparable solutions (TSF, BAHG, DQLTV, GeoQoE, Clone) either do or do not address these challenges?
[265] This literature review is excellent; both well written and well organized. That beings said, it is also extremely dense and might benefit from some subsections as described above. Table 1 is also an excellent summary of the existing body of work and how it relates to your problem. I wonder if there is a way to condense the information in such a way that allows the table to sit on the page a little bit better but I don't have an obvious solution. 
[265] Your first diagram doesn't appear until page 7, and aside from Table 1 the first 7 pages are entirely text. While this text is well-written, I strongly advise including either an additional figure or moving Figure 1 to no later than page 2. As your detailed literature review makes clear, there are MANY solutions to your problem. From the title and abstract alone, I can't fully place your contribution to the larger body of work until I read all the way through to your first figure on page 7. Admittedly, I am not an expert in this particular field, but arguably, neither is your target audience. Including a detailed graphic very early in the manuscript that clearly summarizes your problem and the key components of your solution will do a much better job of getting a readers attention and communicating the key points of your paper without a significant reading investment.
[273] Somebody obviously spent a lot of time and effort creating all the detail in Figure 1; I recommend supporting it with a stronger, more descriptive caption.
[280] Missing space between "percent".
[296-310] This paragraph might warrant an additional block diagram that describes the different divisions, which operations happen in each division, and what information is passed between them.
[361] I would really like to see some sort of flow diagram that summarizes the node selection process that you are describing in Section 3.3.
[424] It looks like your variables are italicized here, but on a previous page they were not. I think either convention is fine but you should pick one and be consistent.
[452-468] You have several lists here, consider separating them into bullet points for clarity. Furthermore, what are the "five main challenges" you reference here? It is not immediately obvious.
[501] Looks like a missing backslash on "alpha".
[510] Sections 3.3, 3.4, and 3.5 would, in my humble opinion, benefit greatly from either a block diagram or a flow diagram that describes the decision and connections that your algorithm makes in the different stages. The text is well-written, but it is a lot of text to read and comprehend which would be significantly more understandable if supported by a diagram(s).
[514] I am not familiar with the NS-3 simulation platform. Is this industry-standard? If so, can you provide a citation so that I might learn more? If not, can you provide some context and its capabilities and characteristics?
[558] Given that you already spent the effort to create Table 1, I would like to see something similar here, but only for the 4 techniques you are using as a comparison. By this point I've already forgotten all of the details about these methods so a small summary table highlighting the key characteristics would be a good refresher before launching into the results.
[610] Nice table, very clean summary of your parameters. Correct the caption that reads "Table 1".
[645] For all of the figures after and including Figure 3, I would like to see the following changes: a) significantly larger font size (labels, axis ticks, legend), b) significantly thicker lines connecting the data points (Figure 3 uses a good line thickness), c) significantly larger markers, d) different colors on the 5 different techniques (given that 3-8, 11, and 13-16 all have the same legend, color coding here will be especially powerful and given that there are only 5 options you can simply use red, orange, green, blue, purple for lack of a better idea). Choose different colors for 9, 10, 12 since the data is different, just choose your three favorite colors and do a gradient for each different plot. The data in these figures is excellent but the overall delivery is severely limited by the lack of the above suggestions.
[819] I would personally like to see a little bit more detail in each of the result figure captions, if nothing more than an additional sentence or two on the key conclusions. Ideally, I should be able to skim this section and get a general idea of your key results without reading the rest of the paper, and I think given the quality of the data this can be achieved with some better presentation (as described in the previous comment) and some more detail in the captions.
[822] I would like to see CNQI redefined in full once again in the conclusion. I love that you include quantitative conclusions here in the conclusion, but I would also like to see a) a few sentences specifically addressing the key challenges you identified in the introduction and how your approach does better, b) even more quantitative conclusions, given that you already went through the effort to produce a wide range of results for numerous different metrics, and c) an overall revision to improve the language of the conclusion. There aren't any particular issues as it stands, but given that most of the manuscript, and the abstract in particular, are well written, I would like to see the same level of language executed in the conclusion.

Thank you for your contributions and for putting together an excellent manuscript. I look forward to your reply and future revisions.

Author Response

Response to Reviewer 2 Comments

Point 1 : [28-32] Since your manuscript is already very long, consider breaking these three key challenges into a numbered list to help break up the significant wall of text.

Response 1 :  Thanks for the detail of the comment. As the reviwer’s comment we changed the format of the introduction with several subsections added.

Point 2 : [121] The first three pages of the document are just a wall of text, please consider adding subsection headers to divide the content like you do in later sections (i.e. Challenges, Background, Contributions, etc.).

Response 2 : As the reviwer’s comment, we separated mentioned section into several subsections to improve the readability.

Point 3 : [33-83] This literature review is well written, well organized, and well cited. My only suggestion is again to separate it from the rest of the text with a subsection header. Nice work here nonetheless.

Response 3 : As we mentioned the comment above, we separated the original paper into several subsections to improve the readability. Thanks for the detail about the comment.

Point 4 : [84-115] You provide a well-written, detailed explanation of your proposed Clone solution. Since you went through the effort of identifying the 3 critical challenges earlier in the introduction, can you add a short paragraph describing how the comparable solutions (TSF, BAHG, DQLTV, GeoQoE, Clone) either do or do not address these challenges?

Response 4 : For the reviwer’s comment response, we have table 1 to explain the summary of other protocols. Our manuscript is already full of text, so we rewrite the explanation of the related work to point out table 1.

Point 5 : [265] This literature review is excellent; both well written and well organized. That beings said, it is also extremely dense and might benefit from some subsections as described above. Table 1 is also an excellent summary of the existing body of work and how it relates to your problem. I wonder if there is a way to condense the information in such a way that allows the table to sit on the page a little bit better but I don't have an obvious solution.

Response 5 : Thanks for the compliment on the summary table. From the format of the manuscript paper. There is no other way to change the table to show better results. We are sorry for the table format.

Point 6 : [265] Your first diagram doesn't appear until page 7, and aside from Table 1 the first 7 pages are entirely text. While this text is well-written, I strongly advise including either an additional figure or moving Figure 1 to no later than page 2. As your detailed literature review makes clear, there are MANY solutions to your problem. From the title and abstract alone, I can't fully place your contribution to the larger body of work until I read all the way through to your first figure on page 7. Admittedly, I am not an expert in this particular field, but arguably, neither is your target audience. Including a detailed graphic very early in the manuscript that clearly summarizes your problem and the key components of your solution will do a much better job of getting a readers attention and communicating the key points of your paper without a significant reading investment.

Response 6 : As the reviwer’s comment. We realized the early of this paper is full of text without any figures and graphs. We rewrite the introduction and the related work to remove the duplicated text. Also, we separated the subsections to improve the readability of the paper as the reviwer’s comment.

Point 7 : [273] Somebody obviously spent a lot of time and effort creating all the detail in Figure 1; I recommend supporting it with a stronger, more descriptive caption.

Response 7 : As the reviwer’s comment, we write the additional explanation of figure 1 in the early section 3. Also, we added the flow diagram of the overall proposed protocol in figure 3 to support the explanation of our work.

Point 8 : [280] Missing space between "percent".

Response 8 : We are really sorry for the mistakes. As the reviwer’s comment, we modify our typos in the overall paper.

Point 9 : [296-310] This paragraph might warrant an additional block diagram that describes the different divisions, which operations happen in each division, and what information is passed between them.

Response 9 : As the reviwer’s comment, we added the flow diagram of the overall proposed protocol at the end of the section 3 to assist the reviwer’s understanding the proposed protocol.

Point 10 : [361] I would really like to see some sort of flow diagram that summarizes the node selection process that you are describing in Section 3.3.

Response 10 : As the reviwer’s comment, we added the flow diagram of the overall proposed protocol at the end of the section 3 to assist the reviwer’s understanding the proposed protocol.

Point 11 : [424] It looks like your variables are italicized here, but on a previous page they were not. I think either convention is fine but you should pick one and be consistent.

Response 11 : Thanks for pointing out our minor typo mistakes. We are sorry for the uncomfortable reading. As the reviwer’s comment, we matched the mentioned point to improve the paper unity of our work. And, we modify our typos in the overall paper.

Point 12 : [452-468] You have several lists here, consider separating them into bullet points for clarity. Furthermore, what are the "five main challenges" you reference here? It is not immediately obvious.

Response 12 : As the reviwer’s comment, we added the flow diagram of the overall proposed protocol at the end of section 3 to assist the reviwer’s understanding. Additionally, we are sorry for the mistakes. We also did typo “five main challenges”. It has to be “Three main challenges,” and we edited the typo paragraph to improve the readability.

Point 13 : [501] Looks like a missing backslash on "alpha".

Response 13 : Thanks for pointing out our typo again. We reviewed our paper carefully to modify the other typos, including the typo the reviewer pointed out.

Point 14 : [510] Sections 3.3, 3.4, and 3.5 would, in my humble opinion, benefit greatly from either a block diagram or a flow diagram that describes the decision and connections that your algorithm makes in the different stages. The text is well-written, but it is a lot of text to read and comprehend which would be significantly more understandable if supported by a diagram(s).

Response 14 : As the reviwer’s comment, we added the flow diagram to assist the understanding of our proposed protocol in time order. In the sections that explain our protocol detail, we edited lack of the information and explanations.

Point 15 : [558] Given that you already spent the effort to create Table 1, I would like to see something similar here, but only for the 4 techniques you are using as a comparison. By this point I've already forgotten all of the details about these methods so a small summary table highlighting the key characteristics would be a good refresher before launching into the results.

Response 15 : Thanks for the comment to improve our paper in detail work. However, our paper has a summary of the comparison work in table 1. Additionally, we explained the comparison protocols in the simulation experiments section. We are a little worried about the duplicate explanation for the 4 comparison protocols. Nevertheless, as the reviwer’s comment, we edited the explanation of the 4 comparison protocols a little to help for reminding the detail.

Point 16 : [610] Nice table, very clean summary of your parameters. Correct the caption that reads "Table 1".

Response 16 : Thanks for pointing out our typos. We reviewed our overall paper to modify the typos.

Point 17 : For all of the figures after and including Figure 3, I would like to see the following changes: a) significantly larger font size (labels, axis ticks, legend), b) significantly thicker lines connecting the data points (Figure 3 uses a good line thickness), c) significantly larger markers, d) different colors on the 5 different techniques (given that 3-8, 11, and 13-16 all have the same legend, color coding here will be especially powerful and given that there are only 5 options you can simply use red, orange, green, blue, purple for lack of a better idea). Choose different colors for 9, 10, 12 since the data is different, just choose your three favorite colors and do a gradient for each different plot. The data in these figures is excellent but the overall delivery is severely limited by the lack of the above suggestions.

Response 17 : As the reviwer’s comment, we edited the mentioned figures and graphs labels, axis ticks, legend, and others to improve the visibility. All format of the previous figures has been edited more significant than before.

Point 18 : [819] I would personally like to see a little bit more detail in each of the result figure captions, if nothing more than an additional sentence or two on the key conclusions. Ideally, I should be able to skim this section and get a general idea of your key results without reading the rest of the paper, and I think given the quality of the data this can be achieved with some better presentation (as described in the previous comment) and some more detail in the captions.

Respose 18 : As the reviwer’s comment, we edited the mentioned figures and graphs labels, axis ticks, legend, and others to improve the visibility. All format of the previous figures has been edited more significant than before. Moreover, we edited the analysis of our experiment results depending on the technical differences and significant reasons among the comparison protocols.

Point 19 : [822] I would like to see CNQI redefined in full once again in the conclusion. I love that you include quantitative conclusions here in the conclusion, but I would also like to see a) a few sentences specifically addressing the key challenges you identified in the introduction and how your approach does better, b) even more quantitative conclusions, given that you already went through the effort to produce a wide range of results for numerous different metrics, and c) an overall revision to improve the language of the conclusion. There aren't any particular issues as it stands, but given that most of the manuscript, and the abstract in particular, are well written, I would like to see the same level of language executed in the conclusion.

Response 19 : As the reviwer’s comment, we rewrite the conclusion section to explain more detail about the previous works and differentiate our proposed protocol. Moreover, we summarized the technology of our proposed protocol to solve the problem of the mentioned key challenges. Also, we rewrite the conclusion with similar flows of the abstract and introduction based on the technology changes to improve the readability of the proposed protocol. Especially, the conclusion contains the information of matching the proposed techniques with the challenging issues mentioned early in this paper. Thanks for the comments to improve our work and give inspiration for future works.

Reviewer 3 Report

The manuscript presents and analyzes a scheme for delivering video over VANETs, using QoS and trajectory prediction for improving the timely and successful delivery of packets. The paper deals with an interesting topic, although there are some issues that should be better addressed by the authors:

1) Reorganize the Introduction and Related Work sections. The Introduction already contains or anticipates features of related works, which are later covered in another Section. Please, state more concisely the problem and move the remaining material to the Related Work Section.

2) Protocol proposal. The proposed scheme is quite complex, overall. But the main problem is that the quantitative design is difficult to understand:

· What is the rationale for the CNQI value/formula? It is actually proportional to the number of users.

· How is the value for H in the CNQI determined?

· What is the meaning of the symbols I equation (3)?

· Same in (4), some symbols do not have a meaning or explanation.

· The synchronization phase is explained with insufficient detail, despite the long text. How does it work, exactly?

It is not possible to assess whether this design is logically correct or not if the above points are not clearly explained.

3) The simulation setup and results are quite comprehensive, but some performance plots require further explanation, in my view:

· In Fig. 4, why is the PDR decreasing with the simulation time? How is the PDR been calculated?

· Similarly in Fig. 6, the PDR decreases with the vehicle speed. Why is this so if the proposed scheme predicts trajectories and takes into account the speed of receivers?

· Figs. 9 and 10 and 12 show that, for a high CNQI and PDR, high values of AP density (around 70%) are necessary.  However, higher AP density implies more communication costs among the nodes in the infrastructure. Has this been measured/quantified?

· Fig 14 shows that performance degrades quite fast as the number of video streams grows. Why is this so? CNQI does not depend directly on the number of video streams, and the same seems to happen with the other formulas used for designing the scheme.

4) A caching mechanism is mentioned several times in the paper, but it is not explained in detail. Please, clarify this point.

5) Prediction of trajectories seems to have been studied by the authors in [24]. What are the novelties here in this work?

Author Response

Point 1 : 1) Reorganize the Introduction and Related Work sections. The Introduction already contains or anticipates features of related works, which are later covered in another Section. Please, state more concisely the problem and move the remaining material to the Related Work Section.

Response 1 : Thanks for the detailed comment on our paper. As the reviwer’s comment we edited the duplicated format of the introduction and related work. We separated into several subsections to improve the readability and reduce the tiredness of reading this paper.

Point 2-1 : 2) Protocol proposal. The proposed scheme is quite complex, overall. But the main problem is that the quantitative design is difficult to understand:

  • What is the rationale for the CNQI value/formula? It is actually proportional to the number of users.

Response 1 : The CNQI score is flexible by topology changes due to several reasons, such as subscriber changes, AP locations, bandwidth changes, and others. All APs and node have CNQI score, which is calculated based on the bandwidth, availability, energy status, and hardware abilities. Moreover, the CNQI score is affected by the number of connected subscribers mainly. In our proposed protocol, CNQI is the basic and the primary technique in the overall protocol. It shows the availability of the APs and nodes, controls the load balance in the topology, and makes it easy to maintain and manage.

Point 2--2 : · How is the value for H in the CNQI determined?

Response 2-2 : H represents the hardware availability factor of the APs and nodes in the proposed protocol. From 0 to 10, we set the all node H factor to 8 in our simulation experiments.

Point 2-3 : What is the meaning of the symbols I equation (3)?

Response 2-3 : We could not find the symbol ‘I’ from the equation (3). However, we found a lack of explanation of all symbols in our work from the explanation of the equations. Thanks for the comment to notice the improvement of our work. We rewrite the equation explanation of symbols of all equations.

Point 2-4 : Same in (4), some symbols do not have a meaning or explanation.

Response 2-4 : We are sorry for missing the explanation of some symbols in all equations. As the reviwer’s comment, we modified and rewrote all the explanations of our paper's symbols and typos.

Point 2-5 :  The synchronization phase is explained with insufficient detail, despite the long text. How does it work, exactly?

Response 2-5 : The synchronization operates after the clone node selections. When the first clone node recognizes the user vehicle is in their communication range, the selected first clone node broadcasts the message exchange packets to the vehicles in the communication range to distinguish the user vehicle that requested the content video data clone has received. After the message exchange has finished, the delay calculation operates to calculate the delay between the clone and the user vehicle to prevent the packet loss due to the delay. Using the delay calculation process results, the clone node runs the mobility calculation task to track the user vehicle to support the seamless connection to prevent the link loss circumstances. The clone node predicts the user vehicle’s mobility based on the vehicle’s trajectory information. Therefore, they can pick another clone node on the user vehicle’s trajectory to prevent link loss. While the user vehicle maintains the link connection with the clone node, the packet calibration task runs to prevent the link loss like the mobility prediction. The mobility prediction and packet calibration run simultaneously to support each other to give high QoS and QoE for the video streaming services. We also added the flow diagram of the overall proposed protocol and rewrote our proposed explanation in section 3 and performance evaluations.

Point 3-1 : The simulation setup and results are quite comprehensive, but some performance plots require further explanation, in my view:

  • In Fig. 4, why is the PDR decreasing with the simulation time? How is the PDR been calculated?

Response 3-1 : PDR has been calculated based on the packet loss rates from the video image file transmission. When the image frame loss has occurred from the data transmission, the loss frame cannot process the image on the screen, and the file error has occurred. In other words, if one image frame loss occurs, there is a missing image frame with the blank after we check all the image files in the transmission. Therefore we can figure out there is a packet loss based on the image frame loss. PDR has been calculated with the mentioned method based on the image frame loss rates from the experiments. If the simulation time goes long, the image loss also increases due to mobility and topology changes. Hence, all the comparison protocol runs the task of the recovering process. However, the simulation results show the different data due to the recovery process changes. Clone runs the retransmission from the clone node because the clone node already received the distributed video file data. It means the one-hop recovery task process runs when the packet loss occurs. However, other protocols have more hop counts and long retransmission paths to recover the packet loss. Even BAHG and TSF are not suitable for video streaming services because they are single packet transmission protocols

Point 3-2 : · Similarly in Fig. 6, the PDR decreases with the vehicle speed. Why is this so if the proposed scheme predicts trajectories and takes into account the speed of receivers?

Response 3-2 : In real-road circumstances, no node has a 100% success rate in data transmission. Due to this reason, we randomly set the success rate of the node from 80 to 90%, depending on the number of the user vehicles. Moreover, the decrement of PDR when the vehicle’s speed changes are the changes of the hop counts for data transmission. Clone has to set more Clone node if the vehicle's speed increases because the vehicle passes through the communication range fastly. Other comparison protocols face much more harsh circumstances. The comparison protocols re-searching the user vehicle number gets increased if the vehicle’s speed gets faster. This problem causes many retransmission tries, and it leads to more packet loss in overall video transmission. Moreover, in the video streaming service, these problem leads to ciritical issues for user QoE and QoS. Therefore, the figure 5 and 6 shows the results of the performance differentiate.

Point 3-3 :  Figs. 9 and 10 and 12 show that, for a high CNQI and PDR, high values of AP density (around 70%) are necessary.  However, higher AP density implies more communication costs among the nodes in the infrastructure. Has this been measured/quantified?

Response 3-3 : In the simulation results of our paper, we do not consider the cost of APs and nodes. In the VANETs, we have enough energy source supply from the vehicle betteries, and RSU also connected the electricity supplier. Also, the most critical performance we have to consider as a great result in the video streaming service is QoS and QoE. Because video streaming services already take a lot of costs and high bandwidth on the topology, showing better QoS and QoE is the most important result in this paper. Hence, the proposed protocol designed to focus on better QoS and QoE than cost efficiency.

Point 3-4 : · Fig 14 shows that performance degrades quite fast as the number of video streams grows. Why is this so? CNQI does not depend directly on the number of video streams, and the same seems to happen with the other formulas used for designing the scheme.

Response 3-4 : In the simulation factors, the number of streams denotes the number of users who want to get the video streaming service in the topology. If the user number gets increased, the overload to the AP also gets increased. Hence it leads to the CNQI decrement. Therefore, if the stream numbers get increased with fixed AP density, the service users get increased, but there is not enough AP has located. As a result, like a figure 14, if the stream numbers get increased, the image loss rates also increase due to the availability of the AP decreasing. To solve this problem, we reduce the overload of AP with increment of AP density on the topology.

Point 4 : 4) A caching mechanism is mentioned several times in the paper, but it is not explained in detail. Please, clarify this point.

Response 4 : In this paper, caching mechanism is applied to the clone node. The distributed video content data is forwarded to the clone node. Moreover, the transmitted video data has to wait until the clone node initiates the data transmission sequence. As the reviewer commented, the mentioned caching mechanism carries the video data in the node until the sequence starts. The proposed protocol focuses on mobility support and video packet transmission with high QoS and QoE. The data caching mechanism in detail is out of our work bound. However, we design the better performance work of our improvement from the caching mechanism for future work. Thanks for the comment

Point 5 : Prediction of trajectories seems to have been studied by the authors in [24]. What are the novelties here in this work?

Response 5 : The protocol mentioned in reference 24 is a single packet transmission protocol in VANETs. Therefore, it is not suitable for video streaming service. As we showed the results of  TSF and BAHG, the single packet transmission protocols have trouble supporting the large video packet transmission size due to the high packet loss while they forward the video file. Moreover, reference 24 only considers the simple energy calculation based on the fixed time and fixed nodes. The differentiates between reference 24 and the proposed protocol are in many ways. Reference 24 is suitable for single-packet transmission using the optimal reception point on the road based on the vehicle’s trajectory information. However, the proposed protocol considers the topology information based on the CNQI scoring method to figure out the topology status. It leads to high QoS and QoE with an excellent load balance. Moreover, the proposed protocol uses a caching mechanism and synchronization process to give the user vehicle a seamless video packet streaming service. It shows the significantly better novelties different than reference 24.

Round 2

Reviewer 3 Report

The authors have provided satisfactory answers and correction to the manuscript which improve its quality and soundness, The paper contains enough contribution and interest to be published.